# Large Language Models for Lossless Image Compression: Next-Pixel Prediction in Language Space is All You Need

**Kecheng Chen**[1]    **Pingping Zhang**[1]    **Hui Liu**[1]    **Jie Liu**[1]
**Yibing Liu**[2]    **Jiaxin Huang**[3]    **Shiqi Wang**[1]    **Hong Yan**[1]    **Haoliang Li**[1*]
[1]City University of Hong Kong [2]Baidu Inc. [3]MBZUAI
cs.ckc96@gmail.com;  haoliang.li@cityu.edu.hk

## Abstract

We have recently witnessed that "Intelligence" and "Compression" are the two sides of the same coin, where the language large model (LLM) with unprecedented intelligence is a general-purpose lossless compressor for various data modalities. This attribute particularly appeals to the lossless image compression community, given the increasing need to compress high-resolution images in the current streaming media era. Consequently, a spontaneous envision emerges: Can the compression performance of the LLM elevate lossless image compression to new heights? However, our findings indicate that the naive application of LLM-based lossless image compressors suffers from a considerable performance gap compared with existing state-of-the-art (SOTA) codecs on common benchmark datasets. In light of this, we are dedicated to fulfilling the unprecedented intelligence (compression) capacity of the LLM for lossless image compression tasks, thereby bridging the gap between theoretical and practical compression performance. Specifically, we propose $P^2$-LLM, a next-pixel prediction-based LLM, which integrates various elaborated insights and methodologies, *e.g.,* pixel-level priors, the in-context ability of LLM, and a pixel-level semantic preservation strategy, to enhance the understanding capacity of pixel sequences for better next-pixel predictions. Extensive experiments on benchmark datasets demonstrate that $P^2$-LLM can beat SOTA classical and learned codecs.

## 1  Introduction

Recently, Delétang et al. (2024) have uncovered that a large language model (LLM), pre-trained on massive text corpora, can achieve competitive lossless compression rates across text, audio, and image modalities. This perspective derives from the so-called philosophy, *"Intelligence" and "Compression" are two sides of the same coin* (MacKay, 2003). Theoretically, minimizing log-loss for next-token prediction in the LLM is equivalent to optimizing a lossless compression objective, positioning the LLM as a *general-purpose* compressor for any modality (Heurtel-Depeiges et al., 2024). This insight is particularly compelling for the lossless image compression community, where the need for more effective compression methods has become increasingly critical in the era of streaming media (Rahman and Hamada, 2019). As advanced LLMs' intelligence gradually outperforms humans in various applications (Hu et al., 2024), a spontaneous envision emerges, *i.e., Can the compression performance of the LLM elevate lossless image compression to new heights?* If the answer is affirmative, the lossless image compression community will

---

*Corresponding author

benefit steadily from the progress in LLM techniques since Huang et al. (2024) revealed that a linear growth relationship between LLM's compression performance and intelligence holds. However, achieving this roadmap is not straightforward. The current LLM-based compressor (Delétang et al., 2024) primarily showcases the methodology and corresponding compression results on grayscale images, leaving it unclear how to extend the lossless image compression capabilities of the LLM to more widely used images, *e.g.,* RGB images. As depicted in Figure 1, the direct application of the existing LLM-based compressor to benchmark datasets comprising RGB images reveals a significant performance disparity compared with state-of-the-art (SOTA) classical lossless codecs.

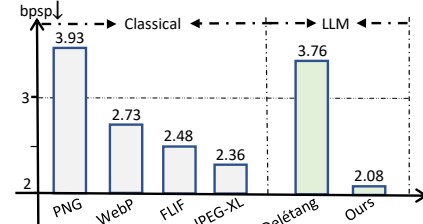

Figure 1: Comparison of different lossless image compressors for bit-per-subpixel (bpsp↓) on CLIC.m dataset. Classical compressors include PNG, WebP, FLIF, and JPEG-XL.

To unlock the potentially unprecedented intelligence (compression) capacity of the LLM and bridge the gap between theoretical and practical compression performance, we carefully analyze three potential limitations of existing LLM-based lossless image compressor (Delétang et al., 2024) when applied to widely-used images. First, compared with next-token (pixel) prediction for grayscale images, modeling the highly nonlinear and long-range correlations of RGB images, where each pixel comprises three subpixels, is more sophisticated. The existing method (Delétang et al., 2024) lacks an effective mechanism to address this challenge. Second, Delétang et al. (2024) impose proxy tokens (*i.e.,* ASCII characters) to represent each pixel value in language space, which would discard the original pixel-level semantic context, impairing the LLM's ability to understand images through their numerical pixel values in language space. Third, Delétang et al. (2024) investigate the general-purpose compression capabilities. Instead, we focus specifically on the image modality, which compels us to develop strategies that enhance the ability for next-pixel predictions.

To address these challenges, we aim to reformulate the overall framework of LLM-based lossless image compression into a new one, which can unlock the inherent intelligence (compression) ability of the LLM to achieve comparative or better compression performance compared with SOTA lossless image codecs. With this goal in mind, our primary motivation is to boost LLM's capacity to comprehend highly complex and long-range correlated pixel sequences in language space, thereby improving next-pixel prediction accuracy, which directly correlates with a better compression ratio (Zhu et al., 2024a). Specifically, we first propose to leverage pixel-level priors (*e.g.,* intra-pixel inter-channel correlation and local self-similarity) and the in-context ability of the LLM to facilitate the understanding of complex RGB pixel sequences. To this end, we integrate these functionalities into a pixel prediction chat template. Second, instead of using proxy tokens, we propose a two-step lossless pixel tokenization strategy that maximizes pixel-level semantic preservation for LLM context understanding, where each subpixel is treated as a "word" that corresponds to a numerical representation in the token dictionary. Finally, we employ a low-rank adaptation (LoRA)-based fine-tuning strategy (Hu et al., 2021), which efficiently and effectively enhances the LLM's understanding capacity of LLM for this customized pixel prediction task. The overall framework is termed as P$^2$-LLM, *i.e.,* next-pixel prediction-based LLM. Our contributions can be summarized as four-fold:

- We aim to fully unlock LLM's unprecedented intelligence (compression) capacity for the lossless image compression task. This perspective bridges the gap between theoretical and practical compression performance for LLM, potentially opening new avenues as LLM intelligence continues to evolve in the future.

- We propose P$^2$-LLM, which integrates various elaborated methodologies, *e.g.,* pixel-level priors, the in-context ability of LLM, and pixel-level semantic preservation strategy. These elements collaboratively enhance the LLM's capacity to comprehend pixel sequences for next-pixel predictions.

- P$^2$-LLM enhances lossless compression rates of LLM-based compressors without extra inference cost. Meanwhile, P$^2$-LLM may be suitable for many offline stream and bandwidth-constrained storage scenarios (*e.g.,* large-scale scientific imaging in astronomy), where data is decoded several months/years after collection in a non-real-time way.

- Extensive experiments demonstrate that (1) Although P$^2$-LLM has no visual-perception architecture, it can achieve competitive performance compared with existing learned codecs

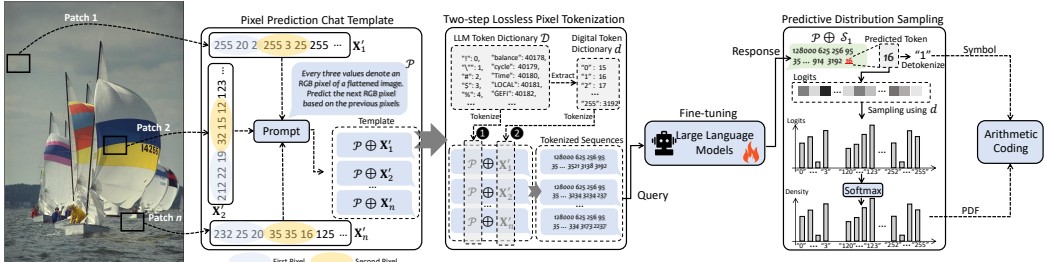

Figure 2: The framework of the proposed P²-LLM, including Pixel Prediction Chat Template for the pixel-level priors and in-context integration in sec. 3.1, Two-step Lossless Pixel Tokenization for pixel-level semantic preservation in sec. 3.2, Predictive Distribution Sampling for scalable probability representation of encoded symbols in sec. 3.3, and Fine-tuning to boost the understanding capacity of pixel sequences in sec. 3.4. You may zoom in for a better view.

for in-domain datasets. (2) P²-LLM exhibits significantly better cross-domain generalization capacity for out-of-distribution datasets compared with classical and learned codes.

## 2 Related works

**Lossless Image Compression.** Traditional lossless image codecs, such as PNG (Boutell, 1997) and JPEG-XL (Alakuijala et al., 2019), operate by employing manual pipelines to diminish the redundancy of images for compression. However, due to the optimized difficulty of traditional codes, the performance gradually bounds with little increase. Thus, the learned image compression (LIC) approaches aim to mitigate such issues by an end-to-end learning framework (Bai et al., 2024). Usually, LIC methods encompass two steps, including 1) statistical modeling of a given image using deep generative models and 2) encoding the given image into the bitstreams using arithmetic coding. Herein, modeling accurate and generalizable statistics of the given image is the key component, where various generative models are used as follows. 1) Autoregressive models, such as PixelRNN and PixelCNN (Van Den Oord et al., 2016), which forecast pixel distributions based on conditional dependencies with previously acquired pixels via masked convolutions. 2) Flow models, *e.g.,* iVPF (Zhang et al., 2021b) and iFlow (Zhang et al., 2021a), leverage invertible transforms to simplify latent distributions for efficient entropy coding. 3) Variational Auto-Encoder models, such as L3C (Mentzer et al., 2019), which utilize variational architectures to model image distributions.

LLM-based compressors belong to autoregressive models, but there are no visual-perception components (*e.g.,* masked convolutions). Instead of perceiving images directly, LLM-based compressors model the statistics of the image in the language space by discretizing each pixel to language tokens.

**Large Language Models for Compression.** The large language model (LLM) has performed surprisingly well in natural language processing (Wu et al., 2023) and computer vision tasks (Yao et al., 2024), due to its accurate next-token prediction capacity. For example, many challenging applications, *e.g.,* machine translation (Feng et al., 2024) and language understanding (Jiang and Li, 2024), are intensively solved by LLM.

Recently, Delétang et al. (2024) demonstrated that language modeling is compression, as log-loss minimization for the next-token prediction of LLM is equivalent to optimizing a lossless compression objective, which enables the LLM as a *general-purpose* compressor for any modality (Heurtel-Depeiges et al., 2024). They showcased that LLM-based compressors can beat some classical codecs (*e.g.,* PNG) for grayscale images. Moreover, recent literature also implies the linear growth relation between compression performance and LLM's intelligence (Huang et al., 2024). These insights motivate the lossless image compression community to investigate the unprecedented intelligence of LLM in more common images, *e.g.,* RGB images. Although it is straightforward to extend Delétang et al. (2024)' approach to RGB images in a channel-independent manner, such a strategy suffers from many limitations. Thus, we are dedicated to fulfilling LLM's unprecedented intelligence (compression) capacity for the lossless image compression task.

# 3 Methodology

**Overall.** From a lossless compression perspective, the proposed $P^2$-LLM (as depicted in Figure 6) aims to render an accurate probability representation of each encoded symbol (*i.e.,* the (sub)pixel) for arithmetic coding. Note that arithmetic coding is acknowledged to be optimal for coding length, where the overall compression performance depends on the abilities of the probabilistic model (Delétang et al., 2024). To this end, we focus on unlocking the unprecedented reasoning capacity of LLM to understand pixel sequence for better next-pixel predictions with accurate probability representations.

## 3.1 Pixel-level Prior and In-context Integration

Assume LLMs can capture implicit structured information and patterns to conduct next-pixel prediction. Such capacity derives from the unprecedented intelligence of the LLM learned from the massive text corpus. However, existing results on benchmark datasets demonstrate that the pre-trained LLM is still behind SOTA codecs with a significant gap.

We argue that this phenomenon may derive from two factors: (1) **Without Pixel-level Priors:** Previous literature realizes accurate next-pixel prediction based on fruitful pixel-level priors,

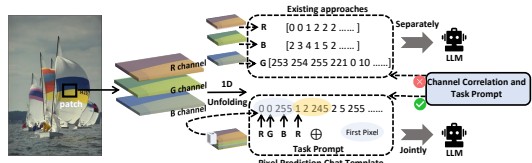

Figure 3: Comparison of different input sequences for pixel prediction. Existing approach is from Delétang et al. (2024). We process each patch of RGB images in a channel-joint manner.

*e.g.,* intra-pixel inter-channel correlation (Van Den Oord et al., 2016; Salimans et al., 2017) and local self-similarity (Zhang et al., 2023; Wewer et al., 2023) between pixels. However, existing LLM-based compressors fail to leverage these pixel-level priors to reason the next prediction, as they either neglect the inter-channel correlation (Li et al., 2024) by channel-independent processing or discard channel information by graying (Delétang et al., 2024). (2) **Without Leveraging In-context Learning of LLMs:** Many works (Dong et al., 2024) demonstrate that in-context learning with well-supported prompts can help the LLM understand specific tasks for more accurate predictions. Existing LLM-based compressors simply input the pixel sequence without motivating potential next-pixel prediction ability under a specific context.

To address the aforementioned limitations, we propose to integrate pixel-level priors and the in-context ability of LLM to enhance the ability to reason the next pixel. To this end, we design a customized pixel prediction chat template to integrate these functions into one. Specifically, given an RGB image $\mathbf{X} \in \mathbb{R}^{W \times H \times 3}$ where $W$ and $H$ denotes the spatial resolution, $\mathbf{X}$ is first flattened into a 1D sequence with $W \times H$ pixels, *i.e.,*

$$\mathbf{X}^{'} = \text{flatten}(\mathbf{X}) = [\mathbf{x}_1, \mathbf{x}_2, \cdots, \mathbf{x}_{W \times H}], \mathbf{x}_i = \{x_i^R, x_i^G, x_i^B\}, \tag{1}$$

where $\mathbf{x}_i$ denotes a pixel that consists of three subpixels $x_i^R, x_i^G, x_i^B$ for red, green, and blue channels. As illustrated in Figure 3, sequential pixels with inter-channel correlation can potentially boost the understanding ability of LLM for spatial relationships, *e.g.,* local self-similarity between continues pixels and color consistency in a specific range. Instead, the relationship between subpixels in a channel-independent manner can be easily disturbed by various interferences, *e.g.,* the noise and the downsampling.

Meanwhile, we introduce a task prompt $\mathcal{P}$ to motivate the in-context understanding ability of LLM. An example task prompt $\mathcal{P}$ can be presented as follows:

> Every three values denote an RGB pixel[a] of a flattened image[b]. Predict the next RGB pixel based on the previous pixels.[c] /////// [a]: Instruct the relationship between sequential values; [b]: Clarify the data format; [c]: Clarify the reasoning task.

We can observe from the task prompt that effective instructions, *e.g.,* the relationship between sequential values and the reasoning task, are provided to potentially enhance the reasoning capacity of LLM for next-pixel predictions.

Formally, our proposed pixel prediction chat template can be represented as $\mathbf{S} = \mathcal{P} \oplus \mathbf{X}'$, where $\oplus$ denotes the concatenation operation. Thus, the conditional probability of a symbol $x_{3i+k}$ ($k$ is 1, 2, and 3 for R, G, and B channel, respectively) given previous $i$ pixels $\mathbf{x}_{1:i} = x_{1:3i}$ and task prompt $\mathcal{P}$ is

$$\rho(x_{3i+k}|x_{1:3i}, \mathcal{P}) = \rho(x_{1:3i+k}|\mathcal{P})/\rho(x_{1:3i}|\mathcal{P}). \tag{2}$$

**Theoretical Analysis.** Different from channel-independent prediction, we use the LLM to conduct a sequential prediction of three channels of a pixel by leveraging inter-channel correlation. To demonstrate the rationality of such a strategy, we first provide the universally-defined prediction theory by neural network-based amortization of Solomonoff induction (Salimans et al., 2017),

**Theorem 1.** *(Li et al., 2024; Grau-Moya et al., 2024) For any parametric meta-learning model $f_\theta$ like the decoder-only large models, if $f_\theta$ is fully trained by log-loss function and consider an infinite sequence $\omega$ of events over a finite alphabet, the optimum posterior distribution $\mu$ of $\omega_{i+1}$ given $\omega_{1:i}$ can be obtained, i.e.,*

$$\lim_{i \to \infty} (f_\theta(\omega_{i+1}|\omega_{1:i}) - \mu(\omega_{i+1}|\omega_{1:i})) = 0. \tag{3}$$

We extend the result in Theorem 1 to our setting, *i.e.,* a sequential prediction of three subpixels of a pixel, to clarify a similar optimum posterior distribution:

**Corollary 1.** *The optimum posterior distribution $\mu$ of $x_{i+1}^R, x_{i+1}^G$, and $x_{i+1}^B$ from a perspective of joint distribution, given previous $i$ pixels $\mathbf{x}_{1:i}$, can be obtained as follows*

$$\lim_{i \to \infty} (f_\theta(x_{i+1}^R, x_{i+1}^G, x_{i+1}^B|\mathbf{x}_{1:i}) - \mu(x_{i+1}^R, x_{i+1}^G, x_{i+1}^B|\mathbf{x}_{1:i})) = 0, \tag{4}$$

*where optimum posterior distribution results in the smallest coding length for arithmetic coding.*

***Proof.*** First, we can decompose the joint distribution $\mu(x_{i+1}^R, x_{i+1}^G, x_{i+1}^B|\mathbf{x}_{1:i}))$ using chain rule:

$$\mu(x_{i+1}^R|\mathbf{x}_{1:i}) \cdot \mu(x_{i+1}^G|\mathbf{x}_{1:i}, x_{i+1}^R) \cdot \mu(x_{i+1}^B|\mathbf{x}_{1:i}, x_{i+1}^R, x_{i+1}^G). \tag{5}$$

Similar decomposition can be conducted by $f_\theta$. By recalling Theorem 1, if each pair of conditional distribution converges independently, a fully trained $f_\theta$ is a must. This means $f_\theta$ has to capture correlations using previous subpixels of the current pixel and previous pixels as the condition, ensuring accurate predictions. However, such domain-specific capacity cannot be guaranteed strictly by pre-trained $f_\theta$. Thus, we assume that such a fully trained decoder-only model can be obtained by fine-tuning pre-trained $f_\theta$ to $\hat{f}_\theta$. Then,

$$\lim_{i \to \infty} (\hat{f}_\theta(x_{i+1}^R|\mathbf{x}_{1:i}) - \mu(x_{i+1}^R|\mathbf{x}_{1:i}) = 0 \tag{6}$$

$$\lim_{i \to \infty} (\hat{f}_\theta(x_{i+1}^G|\mathbf{x}_{1:i}, x_{i+1}^R) - \mu(x_{i+1}^G|\mathbf{x}_{1:i}, x_{i+1}^R)) = 0 \tag{7}$$

$$\lim_{i \to \infty} (\hat{f}_\theta(x_{i+1}^B|\mathbf{x}_{1:i}, x_{i+1}^R, x_{i+1}^G) - \mu(x_{i+1}^B|\mathbf{x}_{1:i}, x_{i+1}^R, x_{i+1}^G)) = 0. \tag{8}$$

As the convergence for each component implies the convergence of the product of these components due to the properties of limits and continuity, Cor. 1 holds. However, fine-tuning $f_\theta$ to model conditional distributions and related correlations is necessary. Otherwise, suboptimal posterior distributions will be due to suboptimal convergence. Proof ends.

Overall, Cor. 1 implies that our channel-joint training can encourage the LLM to implicitly learn a joint distribution over subpixels of a pixel and capture correlations of conditional distributions for optimum posterior distribution. Such modeling will result in more robust and accurate representations, as discussed by Salimans et al. (2017) (which rely on explicitly parameterized modeling). Meanwhile, fine-tuning (as described in sec. 3.4) is indispensable to realize optimum posterior distribution.

## 3.2 Two-step Lossless Pixel Tokenization

The tokenizer is an important component in bridging the original semantic space and discrete language representation used by LLMs. Recent progresses (Ali et al., 2023) highlight the tokenizer choice and corresponding token representations can significantly impact the LLM's downstream performance and reasoning ability. Motivated by this, existing LLM-based compressors may be suboptimal. (1) **Without One-to-One Mapping:** To ensure lossless compression, the tokenizer must enable a one-to-one mapping between the pixel (subpixel for RGB images) value and the token representation.

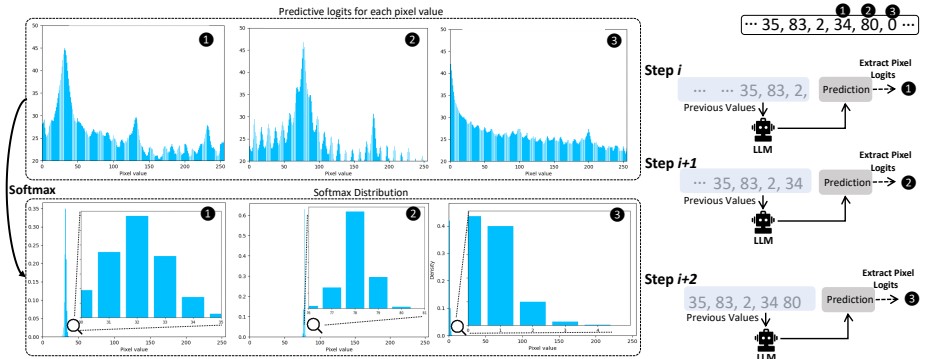

Figure 4: Visualization of predictive distribution sampling for reasoning three subpixels using LLM. Zoom in for a better view.

However, Delétang et al. (2024) conduct a data mapping from the original pixel representation to the range [0,127], as ASCII charterers are encoded in this range. Thus, such preprocessing results in *one-to-two* mapping with information loss. Although they append lost bits to the end of the compressed sequence, the compression ratio decreases. (2)**Without Pixel-level Semantic Context:** Intuitively, if the discrete token representation maintains the original pixel-level semantic context, those pixel-level priors will still be utilized by LLMs. However, existing approaches (Delétang et al., 2024) use the customized tokenizer with ASCII characters but fail to do so, *e.g.,* the pixel values 126, 127, and 0 are represented as "~", "DEL", and "NUL", respectively. The closer relationship between 126 and 127 violates.

To tackle the aforementioned limitations, we propose a two-step lossless pixel tokenization strategy. In a nutshell, our solution derives from the on-hand numerical understanding ability of LLM (*e.g.,* one number is smaller or larger than another) as discussed by Zhu et al. (2024b). Motivated by this finding, we propose to treat each subpixel as a word that corresponds to a numerical representation in the token dictionary, *i.e.,* 127 → "127". By doing so, we can achieve a one-to-one mapping from the pixel value to the token representation, as each digital word from "0" to "255" has a unique token ID. More importantly, due to the preservation of pixel-level semantic context, LLM can understand the sequential pixel values for better next-pixel prediction.

Specifically, the two-step lossless pixel tokenization framework includes 1) a widely used tokenization process and 2) a one-to-one matching process between digital words and digital tokens. Formally, given the tokenization function as $T(\cdot)$ and the corresponding token dictionary as $\mathcal{D}$, we first extract the digital words in the range of 0 to 255 and the corresponding token ID,

$$d = \{\mathtt{str}(z) : T(\mathtt{str}(z), \mathcal{D}) \mid z \in \{0, 1, 2, \dots, 255\}\}, \tag{9}$$

where $d$ denotes the digital token dictionary and $\mathtt{str}(\cdot)$ denotes a number-to-string conversion operator for each pixel value $z$. Furthermore, the task prompt $\mathcal{P}$ and input pixel sequence $\mathbf{X}'$ are tokenized by two steps as follows:

$$\mathbf{S}' = T(\mathcal{P}, \mathcal{D}) \oplus \{d(\mathtt{str}(z_j))\}_{j=1}^{W \times H \times 3}, \tag{10}$$

where we look up the token representation of each pixel using the obtained digital token dictionary $d$, which ensures the one-to-one mapping to avoid potential word splitting using tokenizer $T(\cdot)$.

### 3.3 Predictive Distribution Sampling

LLM can conduct next-token predictions to output the predictive softmax probability. However, only the pixel value prediction is required for lossless image compression. We therefore propose to sample the predictive logits before the softmax layer, based on the digital token dictionary $d$. Then, these sampled logits will be normalized to a probability distribution consisting of 256 probabilities proportional to the exponentials of the input numbers. Such probability distribution can be used for arithmetic coding. Compared with previous works, *e.g.*, Bai et al. (2024) that need to learn parameterized distributions (*e.g.,* Gaussian mixture models), our sampled predictive distribution is parameter-free with maximal scalability and robustness in complex scenarios (Broom et al., 2007).

Formally, given the LLM as a next-token prediction function $\mathtt{LLM}(\cdot)$ *without* the softmax layer, the prediction is represented as $\mathbf{y} = \mathtt{LLM}(\mathbf{S}^{'})$, where $\mathbf{y} \in \mathbb{R}^{1 \times |\mathcal{D}|}$ consists of $|\mathcal{D}|$ predictive logits $\{y_n\}_{n=1}^{|\mathcal{D}|}$ for each token ID, *i.e.*, $n$. By using the digital token dictionary $d$, the pixel-related predictive vector $\mathbf{y}_c$ can be represented as

$$\mathbf{y}_c = \{y_c^z = y_n \mid n = d(\mathtt{str}(z)), z \in \{0, 1, 2, \ldots, 255\}\}, \tag{11}$$

where $\mathbf{y}_c \in \mathbb{R}^{1 \times 256}$. As shown in Figure 4, the predictive logtis between 0 and 255 are presented in the middle column. For each prediction, the peak of those logits is roughly around the encoded pixel value, which showcases that LLM can effectively predict the next pixel. Meanwhile, the visualized logits in Figure 4 demonstrate that LLMs can naturally assign more mass to most possible pixel values. Instead, previous methods (Salimans et al., 2017) need to assign a higher probability to the edge values 0 and 255 in a handcrafted manner.

Finally, the softmax function $\sigma(\cdot)$ is utilized to normalize these logits into a probability distribution, which equals a generalization of the logistic function:

$$p(\mathbf{y}_c) = \sigma(\mathbf{y}_c) = \frac{e^{y_c^z}}{\sum_{z=0}^{255} e^{y_c^z}}. \tag{12}$$

We can use encoded pixel values and probability density functions to conduct arithmetic coding.

### 3.4 Fine-tuning and Practicability

**Fine-tuning.** To enhance the ability of next-pixel prediction of LLM and obtain optimum posterior distribution (as discussed in Cor. 1), it is necessary to fine-tune the LLM using low-rank adaptation (LoRA) (Hu et al., 2021). LoRA enables the LLM to adapt to a customized task in a computationally efficient manner (Li et al., 2023). By following Delétang et al. (2024), we mainly explore the effectiveness of language models for lossless image compression. To this end, the Llama 3 series (Dubey et al., 2024), open-source LLMs released by Meta, are used. We utilize the pre-trained Llama 3 series 8B base model (The effect of other model sizes is presented in Appendix) provided by Huggingface. By following Delétang et al. (2024), we split the overall image into sequential non-overlapped patches for compression. Meanwhile, different patches can be independently processed in a batch manner, which enables parallel acceleration. For the training set, we split 10K and 4,000K patches as the validation and fine-tuning sets, respectively. We set the epoch as 2 and choose the best checkpoint by computing the average cross-entropy loss on the validation set (per 2k iterations). Many LLM-accelerating and GPU-efficient strategies are used for fine-tuning, including DeepSpeed Stage-2, FP16 mixed-precision training, and Flash Attention.

**Practicability of P²-LLM.** Many offline stream and bandwidth-constrained storage scenarios can use the P²-LLM. For example, upon large-scale scientific imaging in astronomy, the massive data may be decoded for months/years after collection in a non-real-time manner. Meanwhile, with the rapid development of LLMs' quantization and inference accelerating, a more lightweight and efficient LLM-based codec with competitive intelligence may be developed for efficient coding.

## 4 Experiments

**Datasets.** Five commonly-used natural image datasets are imposed to evaluate the lossless compression performance of different approaches, including DIV2K validation set, CLIC.p, CLIC.m, Kodak. Meanwhile, two out-of-distribution datasets are used for generalization evaluation, including SCID and BRACS24. We follow Bai et al. (2024) to use the DIV2K high-resolution training dataset (Agustsson and Timofte, 2017) for fine-tuning the LLM, where each image is cropped into non-overlapped patches. The details of adopted datasets can be found in Appendix.

**Training Details.** As the rationale of LoRA is to approximate a large matrix by two low-rank decomposed matrices, the rank and corresponding alpha coefficient in LoRA would significantly affect the performance. We ablate the rank in some predefined values, and the alpha coefficient is twice as much as the rank for a default setting. After ablation analysis (in Appendix), the rank and alpha coefficient are set to 64, and 128, respectively. The target modules of LoRA include query, key, value, and output projections. The patch size determines the length of context information for LLM. We ablate different patch sizes in predefined values and choose the size of $16 \times 16$. The task prompt of LLM is used as described in sec. 3.1 (The effect of other task prompts is presented in Appendix).

Table 1: Lossless image compression performance of different lossless image codecs in terms of bpsp↓. We use official checkpoints provided by L3C and DLPR for testing on SCID and BRACS24. Other results are reported from DLPR's paper. Classical codecs are based on imagecodecs library.

| Category | Codec | In-distribution | | | |
|---|---|---|---|---|---|
| | | DIV2K | CLIC.p | CLIC.m | Kodak |
| Classical | JPEG-XL (Alakuijala et al., 2019) | 2.88 | 2.63 | 2.36 | 2.87 |
| LIC | L3C (Mentzer et al., 2019) | 3.09 | 2.94 | 2.64 | 3.26 |
| | DLPR (Bai et al., 2024) | 2.55 | 2.38 | 2.16 | 2.86 |
| LLM | Delétang et al. (2024) | 4.17 | 3.89 | 3.76 | 3.96 |
| | **P$^2$LLM** | **2.51** | **2.35** | **2.08** | **2.83** |

Table 2: Lossless image compression performance of different lossless image codecs in terms of bpsp↓. We use official checkpoints provided by L3C and DLPR for testing on SCID and BRACS24. Other results are reported from DLPR's paper. Classical codecs are based on imagecodecs library.

| Category | Codec | Venue | In-distribution | | | | Out-of-distribution | |
|---|---|---|---|---|---|---|---|---|
| | | | DIV2K | CLIC.p | CLIC.m | Kodak | SCID | BRACS24 |
| Classical | PNG (Boutell, 1997) | — | 4.23 | 3.93 | 3.93 | 4.35 | 1.79 | 4.99 |
| | JPEG-LS (Weinberger et al., 2000) | TIP-2000 | 2.99 | 2.82 | 2.53 | 3.16 | 2.11 | 4.04 |
| | CALIC (Wu and Memon, 1997) | TIP-1997 | 3.07 | 2.87 | 2.59 | 3.18 | — | — |
| | JPEG2000 (Skodras et al., 2001) | — | 3.12 | 2.93 | 2.71 | 3.19 | 2.15 | 3.83 |
| | WebP (Si and Shen, 2016) | — | 3.11 | 2.90 | 2.73 | 3.18 | 1.24 | 3.94 |
| | BPG (Yee et al., 2017) | — | 3.28 | 3.08 | 2.84 | 3.38 | 1.57 | — |
| | FLIF (Sneyers and Wuille, 2016) | ICIP-2016 | 2.91 | 2.72 | 2.48 | 2.90 | — | — |
| | JPEG-XL (Alakuijala et al., 2019) | — | 2.88 | 2.63 | 2.36 | 2.87 | 1.26 | 3.67 |
| LIC | L3C (Mentzer et al., 2019) | CVPR-2019 | 3.09 | 2.94 | 2.64 | 3.26 | 2.67 | 3.98 |
| | RC (Mentzer et al., 2020) | CVPR-2020 | 3.08 | 2.93 | 2.54 | — | — | — |
| | iVPF (Zhang et al., 2021b) | CVPR-2021 | 2.68 | 2.54 | 2.39 | — | — | — |
| | iFlow (Zhang et al., 2021a) | NeurIPS-2021 | 2.57 | 2.44 | 2.26 | — | — | — |
| | LLICTI (Kamisli, 2023) | TCSVT-2024 | 2.77 | 2.79 | — | 2.99 | — | — |
| | ArIB-BPS (Zhang et al., 2024) | CVPR-2024 | 2.55 | — | — | — | — | — |
| | DLPR (Bai et al., 2024) | TPAMI-2024 | 2.55 | 2.38 | 2.16 | 2.86 | 1.58 | 3.61 |
| LLM | Delétang et al. (2024) | ICLR-2024 | 4.17 | 3.89 | 3.76 | 3.96 | 1.67 | 4.12 |
| | **P$^2$LLM** | This paper | **2.51** | **2.35** | **2.08** | **2.83** | **1.21** | **3.33** |

The initial rate of the cosine decay learning scheduler is set to $1 \times 10^{-4}$ with a warming-up of 1000 steps. We use 4 NVIDIA A800 GPUs for fine-tuning with a batch size of 8 per GPU.

**Baselines.** To evaluate the effectiveness of our proposed method, various baseline codes are introduced as follows: 1) **Classical Codes.** Classical codes usually compress the image using handcrafted priors and elaborate framework designs. Here, we use some widely adopted classical codes, including PNG, JPEG-LS, CALIC, JPEG2000, WebP, BPG, FLIF, and JPEG-XL. 2) **Learned Image Compression (LIC).** LIC models usually directly minimize the rate cost by deep neural networks. In this branch, residual coding-based pipelines have achieved SOTA compression performance, where the residual information of lossy compression is compressed by arithmetic coding. We utilize some SOTA LIC models for comparison, including L3C, RC, iVPF, iFlow, LLICTI, ArIB-BPS, and DLPR. 3) **LLM-based Compressor.** We mainly reproduce Delétang et al. (2024)' method to compress the RGB images by maintaining their key components, including using a pre-trained LLM, proxy tokens, and a channel-independent manner. Practically, we discard the proxy tokens to use our tokenization strategy, as appending lost bits to the end of the compressed sequence is sophisticated. Note that online training-based codecs *e.g.,* NNCP (Bellard, 2021) and CMIX (Knoll et al., 2008) are not compared as all LIC/LLM-based baselines perform offline training and their runtime is huge.

**Main Results.** We evaluate the lossless compression performance of different codes using bit-per-subpixel (bpsp). As illustrated in Table 2, our proposed P$^2$-LLM achieves the best performance in all datasets, compared all classical and LIC baselines with an obvious margin. For example, P$^2$-LLM achieves 2.08 and 2.83 bpsp, suppressing the best LIC approach (DLPR) with 2.16 and 2.86 bpsp. Meanwhile, P$^2$-LLM beats the best classical compressor, JPEG-XL. Note that Delétang et al. (2024) approach only outperforms the PNG, which is reasonable as they cannot generalize to widely-used images (*e.g.,* RGB images) due to the lack of effective pixel-level semantic context and the fine-tuning. Last but not least, P$^2$-LLM exhibits significantly better generalization than learned codes.

Table 3: Ablation Study. Channel-Indep. means RGB images are compressed in a channel-independent manner for LLM. Channel-Corre. means RGB images are compressed by next-pixel prediction, as proposed in sec. 3.1. FT denotes the fine-tuning. w/: With and w/o: Without.

| | Pixel Prediction Chat Template | | | | bpsp↓ |
| | Pixel-level Prior | | In-context Learning | | |
| | Channel-Indep. | Channel-Corre. | w/o Task Prompt | w/ Task Prompt | |
|---|---|---|---|---|---|
| w/o FT | ✓ | ✗ | ✓ | ✗ | 4.55 |
| | ✓ | ✗ | ✗ | ✓ | 4.46 |
| | ✗ | ✓ | ✓ | ✗ | 3.96 |
| | ✗ | ✓ | ✗ | ✓ | 3.83 |
| w/ FT | ✓ | ✗ | ✓ | ✗ | 3.95 |
| | ✓ | ✗ | ✗ | ✓ | 3.76 |
| | ✗ | ✓ | ✓ | ✗ | 2.99 |
| | ✗ | ✓ | ✗ | ✓ | **2.83** |

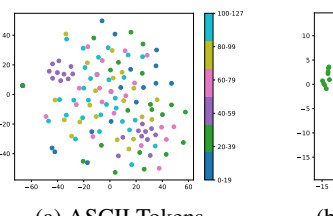

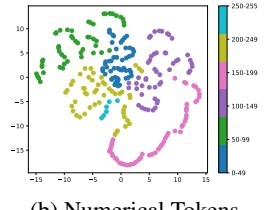

| Codec | Encoding Time | Decoding Time |
|---|---|---|
| JPEG-XL | 0.73 | 0.08 |
| BPG | 2.38 | 0.13 |
| L3C | 8.17 | 7.89 |
| DLPR | 1.26 | 1.80 |
| Delétang et al. (2024) | 15.13 | 272.05 |
| $P^2$-LLM | 14.89 | 273.11 |

| (a) ASCII Tokens | (b) Numerical Tokens | (c) Runtime comparison (second/image) |
|---|---|---|

Figure 5: (a)-(b) Token embedding visualization of pixel values using t-SNE (dimension of each token embedding is 4096 in Llama 3). (a) Pixel values tokenized by proxy tokens as in Delétang et al. (2024) (with data remapping from [0,255] to [0,127]). (b) Using digital token dictionary. (c) Comparison of runtime on Kodak dataset using 8 A800 GPUs (batch size: 16, subprogress: 2 per GPU). $P^2$-LLM and Delétang et al. (2024) adopt the same context, leading to similar runtime.

## 4.1 Detailed Analysis of Each Key Component

We carefully analyze the effectiveness of each key component using the ablation study on the Kodak testing dataset. Some conclusions can be presented as follows.

**– Fine-tuning (as discussed in sec. 3.4) *cannot* fully awake LLM's compression ability.** As illustrated in Table 3, it can be observed that the fine-tuning can significantly improve the compression performance under the same setting, which is reasonable as the LLM's understanding ability increases a lot for pixel sequences, resulting in more accurate next-pixel predictions. However, it should be noted that simply fine-tuning LLM cannot result in SOTA compression performance. For example, a 3.76 bpsp score (last-third row) is achieved with channel-independent and task prompt settings. Such performance still has a significant downside compared SOTA models as in Table 2.

**– Pixel-level priors and In-context learning (as discussed in sec. 3.1) are important catalysts, especially the former.** As illustrated in Table 3, pixel-level priors and in-context learning can improve the compression performance, regardless of with or without fine-tuning settings. Especially, without fine-tuning, we can observe that simple usage of channel correlations can intensively increase the performance, *i.e.,* from **4.55** (first row) to **3.96** (third row) bpsp. This is reasonable as the LLM can leverage the intra-pixel inter-channel correlations for more accurate next-pixel reasoning. Meanwhile, the task prompt can moderately enhance the understanding of LLM for pixel sequence with better compression performance using the context, *e.g.,* from 3.96 (third row) to 3.83 (fourth row) bpsp.

**– Two-step lossless pixel tokenization (as discussed in sec. 3.2) can maintain pixel-level semantic context with more compact representations.** As shown in Figures 5(a) and 5(b), we visualize the token embeddings of pixel values from 0 to 255. These embeddings are queried from the `embed_tokens` layer of LLM. We can observe that the pixels with closer values are roughly closer in feature space when our numerical tokens are adopted, thus pixel-level semantic context can be preserved in language space. This aids LLM in understanding the relationship between pixels better.

**– Predictive distribution sampling (as discussed in sec. 3.3) results in accurate and compact probability representation.** As shown in Figure 4, predictive logits between 0 and 255 are presented in the middle column. For each prediction, the peak of those logits is roughly around the encoded pixel value, which showcases that LLMs can effectively predict the next pixel by understanding

the relationship between pixel values. After the softmax function, the probability representation is extremely compact, which can benefit the AC with better compression performance.

– **Computational Complexity.** From Table 5c, our proposed method has no extra inference cost compared with Delétang et al. (2024). Although the current decoding time is slower than other baselines due to the inherent downside of autoregressive models (Kizhakkumkara Muhamad et al., 2023), we argue that it is feasible to achieve better efficiency with the development of LLM-based inference acceleration and computationally efficient pixel prediction strategies in the future.

## 5 Limitation and Conclusion

Although we have observed that the LLM-based compressor can beat classical and LIC-based codes, especially in its cross-domain generalization capacity, its decoding time is slower than other baselines and similar to autoregressive counterparts. More investigations about balancing effectiveness and efficiency will be explored in the future.

In this paper, to fully utilize LLMs' intelligence for lossless image compression, we introduce $P^2$-LLM to improve lossless image compression performance in the language space. This mitigates the gap between theoretical and practical compression performance for LLM. Extensive experiments show that $P^2$-LLM can beat SOTA classical and learned lossless compressors with obvious gains.

## Acknowledgments and Disclosure of Funding

The research was partially supported by the RGC General Research Fund 11200323, NSFC/RGC JRS Project N_CityU198/24, Hong Kong Innovation and Technology Fund GHP/044/21SZ, and PRP/036/24FX, the Hong Kong Innovation and Technology Commission (InnoHK Project CIMDA), the Institute of Digital Medicine, City University of Hong Kong (Projects 9229503 and 9610034), in part by Chow Sang Sang Donation and Matching Fund (Project 9229161), and in part by the CityU under Project 7006087.

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

Table 4: The datasets for evaluation. The symbols ∗ and † denote fine-tuning and testing datasets, respectively.

| Dataset | Description | # Num. | Avg. Resolution |
|---|---|---|---|
| DIV2K-Training∗ | Natural | 800 | 1080×2048 |
| Kodak† | Natural | 24 | 576×704 |
| DIV2K-Validation† | Natural | 100 | 1080×2048 |
| SCID† | Screen Content | 40 | 720×1080 |
| CLIC.p† | Natural | 41 | 1080×2048 |
| BRACS24† | Medical | 24 | 1526×2897 |
| CLIC.m† | Natural | 61 | 1080×2048 |

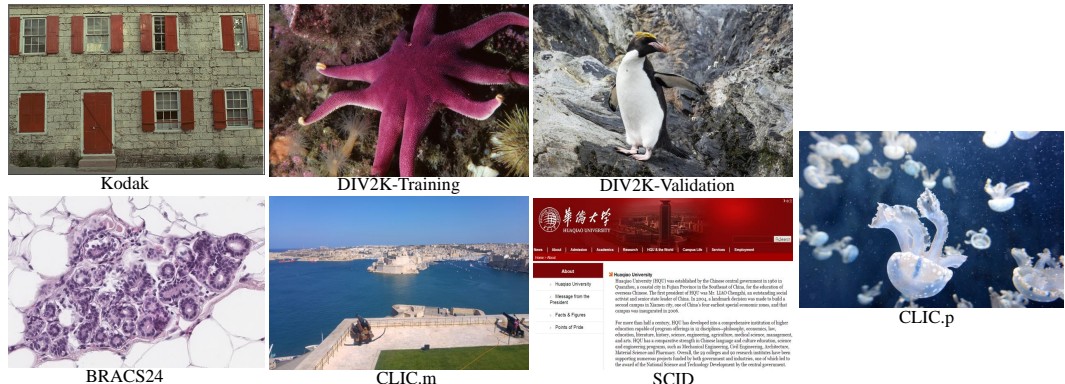

Figure 6: Visualization of used datasets in this paper.

## A Technical Appendices and Supplementary Material

### A.1 Details of Datasets

By following previous works (Bai et al., 2024), seven different datasets with three types of image styles are imposed to evaluate the lossless compression performance of different approaches. As illustrated in Table 1, we follow Bai et al. (2024) to use the DIV2K high-resolution training dataset (Agustsson and Timofte, 2017) for fine-tuning the LLMs, where each image is cropped into around 4,000,000 non-overlapped patches with a size of $16 \times 16$. The testing datasets include Kodak dataset[2], DIV2K high-resolution validation dataset (Agustsson and Timofte, 2017), SCID dataset (Ni et al., 2017), CLIC.p dataset (Toderici et al., 2020), BRACS24 dataset (Niazi et al., 2019), and CLIC.m dataset (Toderici et al., 2020). These datasets have different image styles, including natural images, medical images, and screen content images.

### A.2 Ablation Study of LoRA-based Fine-tuning Hyperparameters

Table 5: Performance comparison of different rank and alpha coefficients on different datasets in terms of bpsp (↓).

| Rank | Alpha | Kodak | DIV2K-Validation | SCID |
|---|---|---|---|---|
| 16 | 32 | **2.83** | 2.54 | 1.23 |
| 32 | 64 | 2.84 | 2.53 | 1.25 |
| 64 | 128 | **2.83** | **2.51** | **1.21** |
| 128 | 256 | 2.87 | **2.51** | 1.25 |

In this section, we aim to investigate the effect of different ranks and alpha coefficients in LoRA-based fine-tuning. As illustrated in Table 5, we can observe that a moderate value in terms of the rank

---

[2]https://r0k.us/graphics/kodak/

and alpha coefficient can achieve a more balanced performance among different datasets. Thus, we choose the rank as 64 and the alpha coefficient as 128.

## A.3 Ablation Study of Different Task Prompts

Every three values denote an RGB pixel[a] of a flattened image[b]. Predict the next RGB pixel based on the previous pixels.[c]

---

[a]Instruct the relationship between sequential values.
[b]Clarify the data format.
[c]Clarify the reasoning task.

(a) Fine-tuning task prompt.

Every three values denote an RGB pixel[a] of a flattened medical image[b]. Predict the next RGB pixel based on the previous pixels.[c]

---

[a]Instruct the relationship between sequential values.
[b]Clarify the data format.
[c]Clarify the reasoning task.

(b) Testing task prompt for medical images.

Every three values denote an RGB pixel[a] of a flattened screen content image[b]. Predict the next RGB pixel based on the previous pixels.[c]

---

[a]Instruct the relationship between sequential values.
[b]Clarify the data format.
[c]Clarify the reasoning task.

(c) Testing task prompt for screen content images.

Table 6: Performance comparison of different task prompts in terms of bpsp ($\downarrow$).

| Dataset | Description | Fine-tuning task prompt | Customized task prompt |
|---|---|---|---|
| BRACS24 | Medical | 0.96 | **0.95** |
| SCID | Screen content | 2.82 | **2.81** |

In this section, we aim to investigate the effect of different task prompts for different datasets (BRACS24 and SCID). To this end, we add the image type of coding image into the fine-tuning task prompt as shown in (b) and (c). We randomly choose an image from these two datasets. As illustrated in Table 6, the customized task prompt can mildly improve the compression performance, which is reasonable as the customized task prompt can enhance the understanding capacity of pixel sequence for LLM using the given context. However, such improvement may be explored by other effective task prompts to do so, in the future.

Table 7: Performance comparison of different model sizes on different datasets in terms of bpsp ($\downarrow$).

| Model | Size | Kodak | CLIC.p | SCID | CLIC.m |
|---|---|---|---|---|---|
| Llama 3.2 base | 1B | 2.87 | 2.41 | 1.25 | 2.15 |
| Llama 3.2 base | 3B | 2.84 | 2.38 | 1.22 | 2.11 |
| Llama 3.1 base | 8B | **2.83** | **2.35** | **1.21** | **2.08** |

Note: Llama 3.2 series is the newest version. Since only 1B and 3B are available for Llama 3.2 series, Llama 3.1 8B is chosen. The slight performance difference of different versions is ignored.

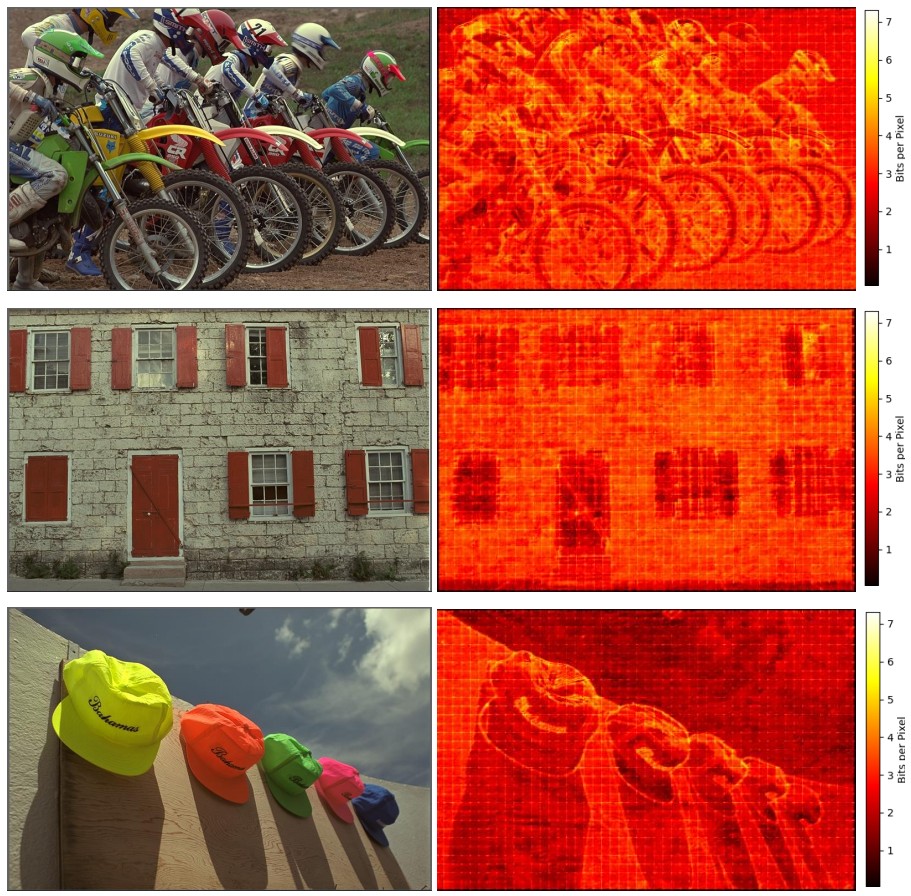

Figure 7: Visualization of bit consumption of example images in Kodak datasets.

## A.4 Ablation study of different sizes of LLM

As illustrated in Table 7, we have observed that the compression performance will significantly benefit from increased model size. This finding would motivate us to investigate more powerful usage of LLMs in the future.

## A.5 Visualization of bit consumption of case images

As we can see, there is less bit consumption in the smooth region with less color change, which is reasonable as the next-pixel prediction is relatively easy in these regions. Instead, in the region of more color change, it is more challenging to predict next pixels, leading to more bit consumption.

