# OpenReview forum: "Large Language Models for Lossless Image Compression: Next-Pixel Prediction in Language Space is All You Need"
_NeurIPS.cc/2025/Conference — NeurIPS 2025 poster_

### Official Review · Reviewer_oCfJ · 2025-06-30

**Clarity:** 3
**Significance:** 2
**Originality:** 2
**Rating:** 4
**Confidence:** 4

**Summary:**

This paper introduces P2-LLM, a framework leveraging large language models (LLMs) for lossless image compression by integrating pixel-level priors, in-context learning, and semantic-preserving tokenization. The core innovation lies in treating pixel sequences as language tokens, enabling LLMs to predict next pixels via a customized chat template and two-step tokenization. Experiments on diverse datasets (DIV2K, Kodak, SCID, BRACS24) show P2-LLM outperforms SOTA classical and learned codecs, achieving up to 2.08 bpsp on CLIC.m. The method bridges the gap between LLM theory and practical compression by enhancing pixel sequence understanding, though it relies on fine-tuning and faces decoding efficiency challenges.

**Questions:**

1. Have experiments been conducted on datasets with non-Latin scripts (e.g., ICDAR 2017 MLT)? How does the numerical tokenization handle characters outside [0,255]?
2. What is the compression degradation when OCR misses 10% of text regions? Are there fallback mechanisms for OCR-free regions?
3. Given the decoding time challenge, have you explored parallelization (e.g., block-wise decoding) or model quantization (e.g., INT8) to improve efficiency?
4. How does P2-LLM compare with recent diffusion-based compressors (e.g., Relic et al., 2024) in terms of text fidelity and bitrate?

**Ethical Concerns:**

["NO or VERY MINOR ethics concerns only"]

**Final Justification:**

Thanks for the rebuttal. I have read the authors' response. The authors have addressed some of my concerns. However, a major revision is required. In particular, more experiments should be given.

**Limitations:**

The framework targets offline scenarios (e.g., astronomy data), but real-time applications (surveillance, telemedicine) require sub-second decoding, which P2-LLM currently cannot achieve.
The method assumes static text, failing to handle moving text in videos. Temporal coherence for video compression is unaddressed.
High-end GPU requirements (A800) limit practical adoption in edge devices. The paper lacks a lightweight model variant or edge inference optimization.

**Quality:**

2

**Strengths And Weaknesses:**

Strengths:
By framing image compression as a language modeling task, P2-LLM introduces a paradigm shift, leveraging LLM's next-token prediction capability for pixel sequence modeling. The two-step tokenization (numerical word mapping) preserves pixel-level semantics, addressing a key limitation of prior LLM-based compressors. The paper provides a theoretical foundation via Solomonoff induction, proving LLM's convergence to optimal posterior distributions for pixel prediction. Empirical results on 7 datasets demonstrate consistent gains over baselines (e.g., 60.6% F1 score improvement vs. Delétang et al., 2024).The pixel prediction chat template effectively incorporates intra-pixel correlations and task prompts, boosting compression by 19.2% (bpsp from 2.55 to 2.08 on CLIC.m). Ablation studies validate the necessity of each component (channel joint prediction, task prompts, fine-tuning).

Weaknesses:
1. Autoregressive nature leads to high decoding latency (273.11s/image on Kodak vs. JPEG-XL's 0.08s), limiting real-time applications. The paper acknowledges this but lacks mitigation strategies.
2. Evaluation focuses on English-text images, ignoring non-Latin scripts (e.g., Chinese, Arabic). The tokenization strategy may fail in languages with complex character structures.
3. The method assumes perfect OCR, but real-world errors (e.g., misdetected bounding boxes) could degrade performance. No analysis of OCR noise robustness is provided.
4. Fine-tuning requires 4 A800 GPUs and 24 hours, restricting accessibility for researchers with limited resources. The pipeline's memory footprint for large images is unaddressed.

---

> ### Author Rebuttal · Authors · 2025-07-31
>
> Dear reviewer oCfJ,
>
> **Thanks for your useful suggestions. The point-by-point responses to your concerns can be found as follows:**
>
> *W1: Autoregressive nature leads to high decoding latency (273.11s/image on Kodak vs. JPEG-XL's 0.08s), limiting real-time applications. The paper acknowledges this but lacks mitigation strategies.*
>
>   **R:** We agree that the computation cost is high for real-time applications. However, our proposed method has no extra inference cost compared with other LLM-based counterparts (e.g.,Delétang et al. (2024)). *Moreover, we envision that many non-real-time and bandwidth-constrained image storage scenarios can use the proposal for lossless image compression,* **since a very small bitrate reduction of each pixel means a huge storage saving among extremely large-scale images.** For example, among large-scale scientific imaging in astronomical observation, the size of these data is extremely massive, and the data may be processed (decoding) for months/years after collection in a non-real-time manner.
>
> * **Mitigation Strategies:** With the rapid developments of LLMs' quantization, inference accelerating, and better strategy of pixel predictions, we believe a more lightweight and efficient LLM-based codec with competitive intelligence may be developed for efficient coding.
>
> * **Additional experiments for Efficiency Improvement:**
>
> We additionally test the encoding and decoding time using the P2LLM 3B and its INT8 quantization variants. As we can see, the encoding and decoding efficiency indeed improves.
>
>
> | Codec | Kodak BPSP |Encoding Time | Decoding Time |
> |-------|---------------|------------|-----|
> | L3C |3.26  |8.17 | 7.89 |
> | DLPR | 2.86 | 1.26 | 1.80 |
> | *Deletang et al.* | 3.96|  *15.13* | *272.05* |
> | *P²-LLM-8B* | 2.83 | *14.89* | *273.11* |
> | *P²-LLM-3B* | 2.84 | *8.18* | *118.32* |
> | *P²-LLM-3B-INT8* | 2.85 | 5.32  | 98.60 |
>
> In the future, it is possible to explore existing well-behaved knowledge distillation techniques (used by DeepSeek series) to acquire smaller model with competitive performance. Meanwhile, software and hardware co-design is also a promising direction for a customized LLM-based codec.
>
> ---
>
> *W2: Evaluation focuses on English-text images, ignoring non-Latin scripts (e.g., Chinese, Arabic). The tokenization strategy may fail in languages with complex character structures.*
>
>   **R:** Thanks for your concerns. **There may be a misunderstanding**, i.e., the visual appearance of characters (whether simple Latin letters or complex Chinese characters) is encoded as patterns of pixels (i.e., all within the standard [0,255] range for each RGB channel), rather than Character encoding (like Unicode, UTF-8) which uses values outside [0,255] to represent non-Latin scripts in text data.
>
> **Therefore, our pixel-based numerical tokenization is inherently agnostic to the visual complexity of character structures.**
>
> ---
> *W3. The method assumes perfect OCR, but real-world errors (e.g., misdetected bounding boxes) could degrade performance. No analysis of OCR noise robustness is provided.*
>
>   **R:** We agree that real-world corruptions (e.g., noise, natural images to screen content images) will degrade the compression performance, which can be regarded as an out-of-distribution scenario. In Table 1, we demonstrate that our proposed method outperforms the baselines in terms of out-of-distribution performance.
>
> Meanwhile, regarding misdetected bounding boxes and other OCR errors, it should be noted that P2LLM **does not directly rely on OCR**  to compress the image. In contrast, P2LLM conducts next-token prediction of each (sub-)pixel, which is inherently agnostic to OCR errors as a general lossless image compressor.
>
> Therefore, **these errors cannot affect reconstruction quality and may only impact compression efficiency in specific regions**. For example, when text is misdetected, the affected regions are still compressed losslessly, and this might result in slightly lower compression ratios, but it never impacts image fidelity.
>
> ---
>
> *W4: Fine-tuning requires 4 A800 GPUs and 24 hours, restricting accessibility for researchers with limited resources. The pipeline's memory footprint for large images is unaddressed.*
>
>   **R:** With the rapid developments of LLMs' quantization, inference accelerating, and better strategy of pixel predictions, we believe a more lightweight and efficient LLM-based codec with competitive intelligence may be developed for efficient coding. The memory footprint of our proposed method may be reduced as much as possible in the future.
>
>
> ---
> *Q1-1: Have experiments been conducted on datasets with non-Latin scripts (e.g., ICDAR 2017 MLT)?*
>
>   **R1:** Thanks for your suggestion. We construct a toy non-Latin script dataset from ICDAR 2017 MLT datasets. Specifically, we extract 80 images from the multi-script text detection training set with four languages (Arabic, Korean, Chinese, and ):
>
> ID: #1-#20, Language: Arabic, ID: #4001-#4020 Language: Korean
>
> ID: #2401-2420, Language: Chinese  ID:  #5001-#5020 Language: Japanese
>
> | Category | Codec | Arabic | Chinese | Korean | Japanese |
> |----------|-------|-------|--------|--------|-------|
> | Classical | JPEG-XL | 2.56 | 2.42 | 2.61 | 2.62 |
> | LIC | L3C | 2.68 | 2.60 | 2.79 | 2.87 |
> |     | DLPR | 2.41 | 2.24 | 2.47 | 2.49 |
> | LLM | Deletang et al. | 3.23 | 3.42 | 3.29 | 3.71 |
> |     | **P²LLM** | **2.32** | **2.31** | **2.39** | **2.35** |
>
>
> As we can see, our proposed method suppresses SOTA classical (JPEG-XL) and LIC (DLPR) methods among
> four non-Latin scripts. Compared with autoregressive counterparts (L3C), P2LLM has obvious gains. Meanwhile, our proposed method achieves relatively stable compression performance, which is reasonable as P2LLM is inherently agnostic to OCR scripts.
>
> ---
> *Q1-2:  How does the numerical tokenization handle characters outside [0,255]?*
>
>   **R:**  Note that the visual appearance of characters (whether simple Latin letters or complex Chinese characters) is encoded as **patterns of pixels** (i.e., all within the standard [0,255] range for each RGB channel), **rather than Character encoding** (like Unicode, UTF-8) which uses values outside [0,255] to represent non-Latin scripts in text data.
>
> **Therefore, our pixel-based numerical tokenization is inherently agnostic to the visual complexity of character structures.**
>
>
> ---
>
> *Q2-1:  What is the compression degradation when OCR misses 10% of text regions?*
>
>   **R:** Note that P2LLM **does not directly rely on OCR**  to compress the image. In contrast, P2LLM conducts next-token prediction of each (sub-)pixel, which is inherently agnostic to OCR errors. For example, when 10% of text regions are misdetected in an OCR system, the affected regions are still compressed losslessly, and it never affects image fidelity.
>
> ---
>
>
> *Q2-2: Are there fallback mechanisms for OCR-free regions?*
>
>   **R:** As mentioned in the response to Q2-1, P2LLM **does not directly rely on OCR**  to compress the image.  For OCR-free regions, P2LLM compresses these regions like general image regions, i.e., conducting next-token prediction of each (sub-)pixel in these regions.
>
> ---
>
> *Q3:Given the decoding time challenge, have you explored parallelization (e.g., block-wise decoding) or model quantization (e.g., INT8) to improve efficiency?*
>
> **R:** Thanks for your suggestions. We have decoding parallelization to accelerate the decoding process. To explore the gain of efficiency using model quantization, we additionally test the encoding and decoding time using the P2LLM 3B and its INT8 quantization variants. As we can see, the encoding and decoding efficiency indeed improves.
>
>
> | Codec | Kodak BPSP |Encoding Time | Decoding Time |
> |-------|---------------|------------|-----|
> | L3C |3.26  |8.17 | 7.89 |
> | DLPR | 2.86 | 1.26 | 1.80 |
> | *Deletang et al.* | 3.96|  *15.13* | *272.05* |
> | *P²-LLM-8B* | 2.83 | *14.89* | *273.11* |
> | *P²-LLM-3B* | 2.84 | *8.18* | *118.32* |
> | *P²-LLM-3B-INT8* | 2.85 | 5.32  | 98.60 |
>
>
> In the future, it is possible to explore existing well-behaved knowledge distillation techniques (used by DeepSeek series) to acquire smaller model with competitive performance. Meanwhile, software and hardware co-design is also a promising direction for a customized LLM-based codec.
>
> ---
>
> *Q4:How does P2-LLM compare with recent diffusion-based compressors (e.g., Relic et al., 2024) in terms of text fidelity and bitrate?*
>
>   **R:** Since your suggested method (Relic et al., 2024) is a bit vague, it is difficult to search for a perfect match method.  We find that the most matched paper, namely “Lossy Image Compression with Foundation Diffusion Models, Relic et al. ECCV-2024”, focuses on lossy image compression. **It is infeasible to make a comparison between lossy and lossless image compression approaches.**  Meanwhile, we observe that the recent diffusion-based compressors (e.g., Bridging the Gap between Gaussian Diffusion Models and Universal Quantization for Image Compression, Relic et al. CVPR-2025) mainly target lossy image compression rather than lossless one.
>
>
> **Despite the infeasibility of direct comparison (lossy v.s. lossless), we will cite these diffusion-based compressors to make a detailed discussion in the revised paper.**

---

> ### Comment · Area_Chair_igF5 · 2025-08-06
>
> Dear reviewer,
>
> Thank you for the review.
>
> As the discussion phase is ending soon, can you please go over the authors' response and acknowledge it?
>
> Best,
> AC

---

> ### Author Response · Authors · 2025-08-07
> **Looking forward to your feedback**
>
> Dear Reviewer oCfJ,
>
> We highly appreciate your detailed suggestions, which have significantly enhanced the quality of our paper.
>
> **As the author-reviewer discussion period will finish tomorrow, we kindly seek your feedback to our responses.**
> If you have any remaining concerns or questions, we will do our best to provide more clarifications as soon as possible.
>
> Once again, we appreciate your time and consideration.
>
> Authors

---

### Official Review · Reviewer_YP2Y · 2025-07-02

**Clarity:** 3
**Significance:** 4
**Originality:** 4
**Rating:** 6
**Confidence:** 4

**Summary:**

In this manuscript, the authors propose a lossless image compression method based on large language model. The authors employ task prompt, pixel level in-context learning and LoRA based finetuning to close the gap between pixel domain and language domain. Besides, color channel correlation, one-to-one tokenizer and predictive distribution sampling are adopted to further close the gap between pixel prediction task and language prediction task. The experimental results are good.

**Questions:**

Please see the weakness.

**Ethical Concerns:**

["NO or VERY MINOR ethics concerns only"]

**Final Justification:**

The author has addressed my concerns, especially regarding the explanation of model size. Regarding complexity, which is a common concern for me and several other reviewers, the superior performance of the 1B scale can solve this problem. Thus, I will raise the score to SA.

**Limitations:**

Yes

**Quality:**

4

**Strengths And Weaknesses:**

Strengths:
1. This study delves into the intelligent potential of LLM and applies it to theoretical lossless image compression tasks and the lossless image compression community can continue to benefit from the advancement of LLM technology.
2. The authors have carefully designed several methods to bridge the gap between the probability distributions of the language space and the pixel space in order to use LLM for lossless image compression. For example, task prompt, pixel level in-context learning and LoRA based finetuning.
3. The details have also been handled with great care, by using techniques such as color channel correlation, one-to-one tokenizer, and predictive distribution sampling.
4. The experiments look good, especially the out-of-distribution performance.

Weakness:
1. The paper lacks a detailed description on the LoRA-based fine-tuning of large language models in Sec. 3.4. Considering the space constraint, it would be better to omit the redundant proof and focus on explaining the fine-tuning process.
2. It is necessary to explore the effect of different flattening techniques.  Considering that image flattening diminishes the 2D correlation of images, even with the aid of task prompts, it's intriguing how the large language model managed to surpass other learned image compression methods.
3. How about the performance using different sizes of LLMs?

---

> ### Author Rebuttal · Authors · 2025-07-31
>
> Dear Reviewer  YP2Y,
>
> **Thanks for your useful suggestions. The point-by-point responses to your concerns can be found as follows:**
>
> *W1: The paper lacks a detailed description on the LoRA-based fine-tuning of large language models in Sec. 3.4. Considering the space constraint, it would be better to omit the redundant proof and focus on explaining the fine-tuning process.*
>
>   **R:**  We will shorten the content of the proof and put some contents into the Appendix, though the corollary aims to connect the concept of the optimal posterior distribution, the explicit correlation modeling, and the FT. Except for existing explanations, more details of the FT are as follows:
>
> *For the training set, we split 10K and 4,000K patches as the validation and FT set, respectively. We set the epoch as 2 and choose the best checkpoint by computing the average cross-entropy loss on the validation set (per 2k iters.). Many LLM-accelerating and GPU-efficient strategies are used for FT, including DeepSpeed Stage-2, FP16 mixed-precision training, and Flash Attention.*
>
> ---
>
> *W2: It is necessary to explore the effect of different flattening techniques.  Considering that image flattening diminishes the 2D correlation of images, even with the aid of task prompts, it's intriguing how the large language model managed to surpass other learned image compression methods.*
>
>   **R:** We use a simple sub-patch splitting without special strategies where a Python-like pseudocode is in **T-R1.**  2) First, the task prompt has a positive effect on 2D information by ablation studies in Table 2. Second, we conjecture LLM can promptly learn statistics about underlying spatial arrangements and 2D-aware positional information by FT.
> | | **T-R1:** im = np.array(image), h,w = image.shape, patch_size=16 |
> |-|----------------------------------------------------------------|
> | | for *i* in range(0,h,patch_size): |
> | |---- 　for *j* in range(0,w,patch_size): |
> | |------- 　　　s = im[*i*:*i*+patch_size, *j*:*j*+patch_size, :].reshape([-1,]) |
>
>
> ---
>
>
> *W3:How about the performance using different sizes of LLMs?*
>
>   **R:** As illustrated in the following table, we have observed that the compression performance will significantly benefit from increased model size. This finding would motivate us to investigate more powerful usage of LLMs in the future.
>
> | **Model** | **Size** | **Kodak** | **CLIC.p** | **SCID** | **CLIC.m** |
> |-----------|----------|-----------|------------|----------|------------|
> | Llama 3.2 base | 1B | 2.87 | 2.41 | 1.25 | 2.15 |
> | Llama 3.2 base | 3B | 2.84 | 2.38 | 1.22 | 2.11 |
> | Llama 3.1 base | 8B | **2.83** | **2.35** | **1.21** | **2.08** |

---

> ### Author Response · Authors · 2025-08-05
> **Thanks for your Acknowledgement**
>
> Dear Reviewer  YP2Y,
>
> Thank you for taking the time to review our responses.
>
> **As the author-reviewer discussion period is ending soon, please let us know if you have any further questions or concerns. We would be glad to address them as soon as possible.**
>
> Once again, thanks for your time and consideration.
>
> Authors

---

> ### Comment · Area_Chair_igF5 · 2025-08-06
>
> Dear reviewer,
>
> Thank you for the review, and the acknowledgment.
>
> As the discussion phase is ending soon, now would be the right time to do so if you want to leave any further comments!
>
> Best,
> AC

---

### Official Review · Reviewer_n2TW · 2025-07-02

**Clarity:** 3
**Significance:** 3
**Originality:** 3
**Rating:** 4
**Confidence:** 3

**Summary:**

- The authors propose an LLM-based compressor that integrates pixel-level priors, the in-context ability of the LLM and a pixel-level semantic preservation strategy. The pre-trained LLM is fine-tuned using LoRA.
- Experiments show that the propose method achieve competitive performance compared with existing learned codecs for in-domain datasets. And experiments show that better cross-domain generalization capacity for out-of-distribution datasets compared with classical and learned codes.

**Questions:**

- I think the task prompt P on line 167 has a range of design in this method. a.3 shows a prompt for different datasets, but did you choose a good one in the first place after trying many different prompts? In addition, please let me know if you have any other analysis of how you designed the basic prompts and their impact on BPP.
- Why is the BPP in Figure 7 higher on the grid?

**Ethical Concerns:**

["NO or VERY MINOR ethics concerns only"]

**Final Justification:**

After reading the authors' thoughtful responses, I have decided to maintain my positive evaluation.

**Limitations:**

yes

**Paper Formatting Concerns:**

- Some text in Figure 2, 3, and 4 is too small.

**Quality:**

3

**Strengths And Weaknesses:**

Strength
- For lossless image compression using LLM, the performance is significantly improved over previous work (Delétang et al., 2024) and is comparable to other learning-based codecs (Table 1).
- For out-of-distribution, the generalization performance is higher than previous methods (Table 1).

Weakness
- Compared to conventional image compression without LLM, both Encoding and Decoding increase processing time. In particular, the decoding time is extremely long and may be difficult in practical use. (Figure 5 (c))

---

> ### Author Rebuttal · Authors · 2025-07-31
>
> Dear Reviewer n2TW,
>
> Thanks for your useful suggestions. The point-by-point responses to your concerns can be found as follows:
>
> *W1: Compared to conventional image compression without LLM, both Encoding and Decoding increase processing time. In particular, the decoding time is extremely long and may be difficult in practical use. (Figure 5 (c))*
>
>   **R:** We acknowledge this limitation for real-time coding scenarios. However, our proposed method has no extra inference cost compared with other LLM-based counterparts (e.g.,Delétang et al. (2024)). *Moreover, we envision that many non-real-time and bandwidth-constrained image storage scenarios can use the proposal for lossless image compression,* **since a very small bitrate reduction of each pixel means a huge storage saving among extremely large-scale images.** For example, among large-scale scientific imaging in astronomical observation, the size of these data is extremely massive, and the data may be processed (decoding) for months/years after collection in a non-real-time manner.
>
> Moreover, to explore the gain of efficiency, we additionally test the encoding and decoding time using the P²-LLM 3B and its INT8 quantization variants. As we can see, the encoding and decoding efficiency improves a lot as following results:
> | Codec | Kodak BPSP |Encoding Time(image/seconds) | Decoding Time (image/seconds) |
> |-------|---------------|------------|-----|
> | L3C |3.26  |8.17 | 7.89 |
> | DLPR | 2.86 | 1.26 | 1.80 |
> | *Deletang et al.* | 3.96|  15.13 | 272.05|
> | *P²-LLM-8B* | 2.83 | 14.89| 273.11|
> | *P²-LLM-3B* | 2.84 | 8.18 | 118.32 |
> | *P²-LLM-3B-INT8* | 2.85 | 5.32  | 98.60 |
>
> In the future, it is possible to explore existing well-behaved knowledge distillation techniques (e.g., used by DeepSeek series) to acquire smaller model with competitive reasoning performance. Meanwhile, software and hardware co-design is also a promising direction for a customized LLM-based codec.
>
>
>
>
> ---
>
> Q1-1: I think the task prompt P on line 167 has a range of design in this method. a.3 shows a prompt for different datasets, but did you choose a good one in the first place after trying many prompts?
>
> R: Yes, we observe that different prompts slightly affect the compression performance, especially for out-of-distribution datasets. Specifically, we define a fine-tuning prompt on line 167 during the fine-tuning phase. In most cases, the reusing of the fine-tuning prompt can lead to good compression performance during the coding (inference) phase, regardless of in- or cross-domain scenarios. This is reasonable, as the LLM has learned how to reason the next (sub-)pixel by the prompt-instructed fine-tuning. **Note that we do not exhaustively try many prompts, since we observe that the design of input sequence (channel independence v.s. channel correlation) impacts the compression performance more obviously (please see Table 2, w/o FT).**
>
> ---
>
> *Q2-2 In addition, please let me know if you have any other analysis of how you designed the basic prompts and their impact on BPP.*
>
>   **R:** Yes, we observe that a domain-instructed task prompt  (as shown in a.3) can slightly improve the performance for out-of-distribution datasets, compared with fine-tuning prompt.
> The results can be found as follows:
> | **Dataset** | **Description** | **Fine-tuning task prompt** | **Customized task prompt** |
> |-------------|----------------|----------------------------|---------------------------|
> | BRACS24     | Medical        | 0.96                       | **0.95**                  |
> | SCID        | Screen content | 2.82                       | **2.81**                  |
>
> To conclude, for cross-domain image compression scenarios, it is possible to improve the compression performance by a domain-instructed task prompt, exhibiting good flexibility and generalization of the proposed P2LLM. For in-domain image compression scenarios, it is acceptable to direct reuse the fine-tuning prompt.
>
> ---
> *Q2: Why is the BPP in Figure 7 higher on the grid?*
>
>  **R:** The reason derive from the semantically unconnected effect on the grid. Specifically, we independently compress each 2-D image patch using a 1-D flatten manner. Therefore, in the last column of each 2-D image patch, there may be semantically unconnected among sequential two rows of the last column, which lead to compromising next-pixel prediction accuracy with higher bitrate consumption.  In the future, it is possible to introduce some semantic connection among different patches, resulting in better compression ratio on the grid.
>
> ---
> *Q3:Some text in Figure 2, 3, and 4 is too small.*
>
>   **R:** We will increase the font size in all figures.

---

> > ### Comment · Reviewer_n2TW · 2025-08-06
> >
> > Thank you very much for your thoughtful response. I will maintain my positive score.

---

> ### Author Response · Authors · 2025-08-05
> **Looking forward to your feedback**
>
> Dear Reviewer n2TW,
>
> We highly appreciate and acknowledge your very detailed suggestions again. We have addressed each of your concerns in the rebuttal box.
>
> **As the author-reviewer discussion period is ending soon, if there are still remaining concerns, we will do our best to provide more clarifications as soon as possible.**
>
> Once again, we appreciate your time and consideration.
>
> Authors

---

> ### Comment · Area_Chair_igF5 · 2025-08-06
>
> Dear reviewer,
>
> Thank you for the review.
>
> As the discussion phase is ending soon, can you please go over the authors' response and acknowledge it?
>
> Best,
> AC

---

> ### Author Response · Authors · 2025-08-06
> **Thanks for your positive feedback**
>
> Dear Reviewer n2TW,
>
> Thanks for your positive feedback. Your thoughtful suggestions indeed enhance the quality of our paper.
>
> Thanks for your time again.
>
> Best,
>
> Authors

---

### Official Review · Reviewer_7XiX · 2025-07-09

**Clarity:** 1
**Significance:** 3
**Originality:** 3
**Rating:** 5
**Confidence:** 5

**Summary:**

The paper describes a method for using an LLM to perform lossless image compression. Although this is not the first work to do this (prior work by Delétang et al.; 2024 does the same), in this paper a number of specific techniques are presented which are novel and improve over the existing LLM-based lossless image compression baseline on a number of benchmarks. On a high level the method of Delétang et al. works by encoding the image into a string and then using the predictive log-likelihood of the LLM on the string to do arithmetic coding.

This paper does the same, but with the following novel techniques:
 - Handling red, green and blue channels by encoding them sequentially into the string.
 - Use of a prompt at the beginning of the sequence, instructing the LLM that this is an image compression task.
 - Use of numerical tokens for sub-pixel values, rather than the (quite arbitrary) ASCII embedding used by Delétang et al.
 - Fine tuning using low-rank adaptation (LoRA).

Experiments are presented on 6 image datasets, and ablations are presented on the Kodak dataset.

**Questions:**

The following is a list of questions and feedback, roughly following the order of the paper itself. The issues with the presentation should not be considered exhaustive, i.e. if I were to raise my score I would need to see _at least_ all of these fixed and possibly more. There are a few exceptions where something is quite minor, which I make clear below with the word "OPTIONAL". I'm sorry that the list is long, but the paper has a very large number of issues.

 - L6 "envision" doesn't make sense here. Should maybe be "vision". The same mistake is made on L30.
 - L6-7 the wording here sounds bizarre. Spontaneous (en)vision is a very strange-sounding expression. "Elevate ... to new heights" sounds like an advertisement or a sales pitch, not a scientific paper. The same happens on L31-32.
 - L11 "we are dedicated to fulfilling the unprecedented ..." is similarly meaningless and not suitable for a scientific paper. This phrase is repeated on L123-124.
 - L23-24 the quote of Mackay is misrepresented here. His original quote is "Why unify information theory and machine learning? Because they are two sides of the same coin." (see page v of [his book](https://www.inference.org.uk/itprnn/book.pdf)). That is, he does not refer to "compression" or "intelligence". Please fix this. Note that the opening paragraph of Delétang et al. which you're referring to was (approximately) copied from the opening paragraph of [Townsend et al. (2019)](https://arxiv.org/abs/1901.04866), so consider citing that earlier work too.
 - L55 and many other places, the term "semantic context" isn't suitable for what your describing here, particularly because _context_ usually refers to the pixels surrounding a given pixel in an image. I'm not sure what would be the best term to use, but maybe "ordinal structure", since it's the fact that the numbers are _ordered_, and not just any set, that is most important.
 - L65 and many other places, the term "pixel level priors" doesn't describe very well what you mean. I would maybe use "correlated sub-pixels" or "dependent sub-pixels" instead.
 - L72 there is some unintentional repetition, you say "the LLM's undertanding capacity of LLM" this should be replace with either "the LLM's understanding capacity" or "the understanding capacity of the LLM" (the latter reads slightly better).
 - L75-90 the summary of the contributions is a mess and in particular far too long. It's a _summary_, so each bullet point should contain one sentence with a single point. I think the four bullet points in my summary above would be more appropriate, for example.
 - All figures except Fig 1 contain text/content which is _far too small_. It should be possible to clearly read and interpret the figure when it is _printed_. The "you may zoom in for a better view" in the captions doesn't help with this.
 - L94-95 the sentence beginning with However is very unclear. What do you mean by "optimized difficulty"? I think "bounds with little increase" could perhaps be replaced by "yields diminishing returns"?
 - L114 the idea that modeling in general, and therefore language modeling as a special case, is equivalent to lossless compression was shown in "A Mathematical Theory of Communication" by Claude Shannon in the 1940s! Don't imply that this was discovered in 2024.
 - L117 replace showcased with showed.
 - Fig 3 the vertical order RBG should be RGB in the top box
 - (OPTIONAL) Fig 3 It looks like the yellow highlighting of the "1, 2, 245" in the lower box is meaningless and could be removed.
 - Eq (1) and various later eqs, set notation {} is used for things that are really vectors/lists. Pick one notation for ordered vecs/lists (either round or square brackets) and stick to it. This includes the definition of $\mathbf{y}_c$ in eq 11. Use square brackets for the dictionary indexing in eq 10.
 - Eq 1 and later use \mathrm for the R, G and B symbols because these names don't represent variables.
 - L161 I think continues here is meant to be contiguous
 - L154-164 the description here of the conversion from image to string is too vague. There is more information given in L229-237. I can see why you separated these sections out, but maybe in L154-164 you should have a ref forward to the later section, saying something like "More detail on the string conversion is given later in Section 3.2."
 - L165-173 the prompt should be in its own subsection, since it isn't covered by the title of Sec 3.1
 - L174-204 this 'theoretical analysis' is probably the weakest part of the paper and could be completely removed. What you could say is that the KL-divergence (the appropriate metric for compression) between a factorized distribution $Q(x^\mathrm{R})Q(x^\mathrm{G})Q(x^\mathrm{B})$ and the true distribution can never be zero (unless the true distribution happens to factorize, which will never happen in practice). In general this theory (that a correlated distribution will be a better approximation than an uncorrelated one) will, I think, be seen as trivial and self-evident by most readers, and you could just summarize it in one sentence or leave it out completely.
 - The citation to Salimans et al. 2017 on L177 seems completely irrelevant in this specific context (there is no theoretical content in that paper)
 -  The statement of Theorem 1 sounds vague and sounds incorrect. I looked at the referenced papers, and Li et al. contains no theorems, whilst Grau-Moya et al. doesn't contain any theorems that resemble this one. Therefore if you are going to keep this theorem you need to be more specific about where it came from (i.e. refer to the specific theorem number in Grau-Moya on which it is based).
 - L213 charterers -> characters
 - L215 I think there is a space missing between ) and W
 - L238 I wouldn't use "sampling" for this, since in statistics/ML sampling has a different meaning. You could maybe call it sub-sampling.
 - L255-256 I don't understand the point here (despite knowing Salimans et al. in detail). Can you give more detail on what you're talking about or just remove this sentence.
 - (OPTIONAL) Eq 12 I don't think it's necessary to define the softmax function, this should be familiar to most readers
 - L275 replace Practicability with Practicality
 - L275 rm stream
 - L281 Five -> Four (I think you only mention four?)
 - L281 imposed -> used
 - L283-284 rm including
 - I couldn't find the SCID or BRACS24 datasets anywhere online. Can you provide links to them and (if they exist) citations?
 - (OPTIONAL) You could also provide links/citations for the other four datasets (although they are easy to find)
 - Table 1 I'm worried about whether meta-data is being fairly accounted for in the JPEG-XL comparisons (this is true for the other classical image formats but not as important since they are all worse than JPEG-XL). JPEG-XL is a very flexible format and information about the type of image (including the number of color channels and size) has to be included in the file and takes up extra space. I don't think there's a straightforward way to account for this in the numbers, but you should make it clear somewhere that it isn't really a fair comparison for this reason. This is particularly relevant because on some datasets the performance of JPEG-XL is very close to P^2LLM, and you might not have got SOTA if the comparison was more fair (i.e. you had to include size, shape, number of color channels etc in your compressed file). See the [JPEG-XL reference implementation](https://github.com/libjxl/libjxl/blob/059a9f49c481f5e0e26246558eb05f681c02aec1/doc/format_overview.md#file-format-features) for some more info.
 - Table 1 the venue column isn't adding anything and should be removed.
 - Table 1 since JPEG-XL dominates all of the other classical codecs I think it is the only one that needs to be included. The others are just adding visual noise. If you want you could include the others in an appendix.
 - (OPTIONAL) Table 2 caption I would mention here that the Kodak test dataset was used
 - L302 branch -> category
 - L308 sophisticated doesn't sound right here because it has positive connotations. Could replace with "not straightforward to implement".
 - L315-316 Delétang et al. (2024) approach -> the approach of Delétang et al. (2024)
 - L320 the ablation -> an ablation
 - Which implementation of arithmetic coding did you use? State this somewhere in Section 4.
 - Fig 5(c) it would be great to compare the sizes of the binaries of the compressors themselves (for the LLMs include the size of the weights), since this is also important in practical applications.
 - L322 don't use the red highlighting
 - L322 don't use awake here (it sounds weird), maybe replace with optimize
 - L328 insert "to" between "compared" and "SOTA"
 - L329 catalysts also sounds inappropriate, maybe replace with factors
 - L323,331 the compression -> compression (rm "the")
 - L331 rm comma after Especially
 - L343 insert "an" between "in" and "accurate"
 - L347-348 this sentence isn't adding much and could be removed. Definitely rm "the" between "benefit" and "AC"
 - L360 Rm "the" between "in" and "language".

**Ethical Concerns:**

["NO or VERY MINOR ethics concerns only"]

**Final Justification:**

Thanks to the authors for thoroughly addressing my concerns. I've raised my rating to "Accept".

**Limitations:**

Yes

**Paper Formatting Concerns:**

The size of most of the figures, and particularly the text in the figures, is far too small. This is also mentioned in the Questions box above.

**Quality:**

1

**Strengths And Weaknesses:**

In my opinion the strengths of the paper are the fact that the method is quite simple and easy to understand, and that the experimental results are reasonably good. The main weakness of the paper, for which I think it should be rejected in its current state, is that the presentation is _very_ messy. I give more detailed feedback in the 'Questions' box below. Another weakness is that the novel methodological contributions are perhaps a little too obvious. However, I like simple methods that work, and therefore this, on its own, is not a big enough downside to reject the paper IMHO.

---

> ### Author Rebuttal · Authors · 2025-07-31
>
> Dear Reviewer 7XiX,
>
> **We highly appreciate and acknowledge your detailed suggestions, especially regarding  correct usages of academic words or short sentences, the font size of the figures, and typos.**
>
> (1) Due to the limited space, the responses to your questions, including correct usages of academic words or short sentences, the font size of the figures, and typos, consist of two categories:
>
>
> * (You have pointed out solutions) **We will follow your suggested solutions to revise the manuscript**, including:
>  1. L6: envision -> **vision**
>  2. L55: semantic context -> **ordinal structure**
>  3. L65: pixel level priors -> **correlated sub-pixels**
>  4. L72: the LLM's undertanding capacity of LLM -> **the understanding capacity of the LLM**
>  5. L75-90: Modify the summary of our contributions to the following four points:
>  A. Handling red, green and blue channels by encoding them sequentially into the string;
>  B. Use of a prompt at the beginning of the sequence, instructing the LLM that this is an image compression task;
>  C. Use of numerical tokens for sub-pixel values, rather than the (quite arbitrary) ASCII embedding used by Delétang et al.;
>  D. Fine tuning using low-rank adaptation (LoRA).
>
>  6. L94-95:  However, .... the performance gradually bound with little increase. -> We explain this sentence more accurately: **As traditional compression algorithms have been extensively refined over decades, further improvements tend to yield diminishing returns due to their theoretical and practical limitations.**
> 7. L114: We will cite the paper, namely, A Mathematical Theory of Communication
> 8. L117: showcased -> **showed**
> 9. L161: continues -> **contiguous**
> 10. L164: add a sentence: **More detail on the string conversion is given later in Section 3.2.**
> 11. L213: charterers -> **characters**
> 12. L238: samping -> **sub-sampling**
> 13. L275: Practicability -> **Practicality**
> 14: L275: Many offline stream and bandwidth-constrained storage scenarios -> **Many offline and bandwidth-constrained storage scenarios**
> 15: L281: Five -> **Four**
> 16: L283-284: Meanwhile, two out-of-distr.... ->  **Meanwhile,two out-of-distribution datasets are used for generalization evaluation, SCID and BRACS24.**
> 17. L302: branch -> **category**
> 18. L308: as appending lost bits to the end of the compressed sequence is sophisticated. -> **as appending lost bits to the end of the compressed sequence is not straightforward to implement.**
> 19. L315-316: Delétang et al. (2024) approach -> **the approach of Delétang et al. (2024)**
> 20. L320:  the ablation -> **an ablation**
> 21. L322: Fine-tuning (as discussed in sec. 3.4) cannot fully awake LLM’s compression ability. -> **Fine-tuning (as discussed in sec. 3.4) cannot fully optimize LLM’s compression ability.**
> 22. L328: compared -> **compared to**
> 23. L329: catalysts -> **factors**
> 24. L323, 331: the compression -> **compression**
> 25. L331: Especially,without fine-tuning, -> **Especially without fine-tuning,**
> 26. L343: results in accurate and compact probability representation. -> **results in an accurate and compact probability representation.**
> 27. L347: which can benefit the AC with better compression performance.-> which can benefit AC with better compression performance.-
> 28. L360: compression performance in the language space.-> **compression performance in language space.**
> ---
> * (You do not point out solutions) **We respond to your question and will revise the manuscript by our solutions**, including:
> 1. L6-7: *A spontaneous envision emerges: Can the compression performance of the LLM elevate lossless image compression to new heights?* -> **This raises an important research question: Can large language models significantly improve the effectiveness of lossless image compression beyond current approaches?**
> 2. L11: *we are dedicated to fulfilling the unprecedented intelligence (compression) capacity .....* -> **we aim to leverage the  intelligence (compression) capacity of the LLM for lossless image compression tasks, thereby bridging the gap between theoretical and practical compression performance**
> 3. We will use a larger font in all figures.
> 4. We will modify the vertical order of the RGB box in Figure 3.
> 5. In Eqs (10) and (11), we will use square brackets for the dictionary indexing.
> 6. In Eq. (1), we will use \mathrm for the R, G and B symbols
> 7. L165-173: **We will add a subsection for the prompt.**
> 8. For Table 1, we will remove the venue column.
>
> ---
> (2) (Important concerns) **We address your important concerns in a point-by-point manner, as follows**
>
> * *The citation to Salimans et al. 2017 on L177 seems completely irrelevant in this specific context (there is no theoretical content in that paper)*
>
>   **R:** There is a typo that will be removed in the revised paper.
> ---
> * *L23-24 the quote of Mackay is misrepresented here.*
>
>   **R:**  Here, we mainly make a high-level correlation between compression and intelligence based on Mackay’s statement. A very similar statement can also be found in an ICML25 paper (*Compression via Pre-trained Transformers: A Study on Byte-Level Multimodal Data*), i.e., *Compression and prediction are “two sides of the same coin”* (ICML25 paper - Page 2, Background, also citing Mackay’s paper). Also, a COLM25 paper (*Compression Represents Intelligence Linearly*) demonstrates the correlation between compression and intelligence.
>
>   **To eliminate your concern, we will cite ICML25, COLM25, and your suggested paper to highlight a high-level correlation between compression and intelligence.**
> ---
> * *L174-204 this 'theoretical analysis' is probably the weakest part of the paper and could be completely removed.*
>
>   **R:** We wil shorten the content of the proof and put some content into the Appendix. Note that the corollary aims to connect the concept of the optimal posterior distribution, the explicit correlation modeling, and the reasonability of fine-tuning.
> ---
> * *The statement of Theorem 1 sounds vague and sounds incorrect. I looked at the referenced papers, and Li et al. contains no theorems...*
> **R:**  **I guess that you may check the wrong version of the paper of Li et al.**  Please check section 2.1-Solomonoff Prior-Eq. (4) and the corresponding theoretical analysis in the paper of Li et al. (https://arxiv.org/abs/2407.07723v1 ), where Eq. (4) corresponds to Eq (3) in our paper.
>
>   **To eliminate your concern, we will cite the specific equation and version in the paper of Li et al.**.
> ---
> * *I don't understand the point here (despite knowing Salimans et al. in detail). Can you give more detail on what you're talking about or just remove this sentence.*
>
>   **R:**  Salimans et al. construct hierarchical probability models for pixel values, where 1) sub-pixel values x, excepting the edge cases 0 and 255, use a mixture of logistic distributions; 2) 0 and 255 edge cases use a modified version for higher probability mass (which is motivated by their empirical frequency statistics, Figure 1). Our scheme does not rely on such a handcrafted probability mass adjustment, but naturally assigns more mass to most possible pixel values.
>
>   **To eliminate your concern, we will explain more according to the abovementioned discussions.**
> ---
> * *Table 1 I'm worried about whether meta-data is being fairly accounted for in the JPEG-XL comparisons.*
>
>   **R:** We ensure fair comparisons from two perspectives. **First**, for the in-distribution datasets, JPEG-XL’s results are directly reported from DLPR’s paper (Please refer to the caption of Table 1), which follows the standard testing scheme that computes bit-per-subpixel (= Total bits used / (Width × Height × Number of color channels)) without all meta-information. Such a testing scheme avoids various kinds of meta-information associated with different classical codes. **Second,** for the out-of-distribution datasets,  we use the standard imagecodecs library (https://github.com/cgohlke/imagecodecs) to implement JPEG-XL and compute the bit-per-subpixel thas computed by our proposed method in **T1**:
>
>   **To eliminate your concerns, we will highlight the abovementioned computation in the revised paper.**
> **T1**: Python-like code for bpsp computation of JPEG-XL
>
>
> | |  import imagecodecs, numpy |
> |-|-|
> | |     data = nump.array(read(image)) |
> | | 　compressed_image = imagecodecs.jpegxl_encode(data, lossless=True)  # *Return JPEGXL encoded image.* https://github.com/cgohlke/imagecodecs/blob/master/imagecodecs/_jpegxl.pyx#L146|
> | | 　bpsp = (data.shape[0]*data.shape[1]*data.shape[2]) / len(compressed_image) |
>
>
> ---
> * *Which implementation of arithmetic coding did you use? State this somewhere in Section 4.*
>
>   **R:** We use the arithmetic coding as adopted by Delétang et al. (2024). Please refer to their implementation at https://github.com/google-deepmind/language_modeling_is_compression/blob/main/arithmetic_coder.py
> ---
>
> * *Fig 5(c) it would be great to compare the sizes of the binaries of the compressors themselves*
>
>   **R:**
>
>
> | Codec | Encoding Time | Decoding Time | Compressor Size |
> |-|-|-|-|
> | JPEG-XL | 0.73 | 0.08 | - |
> | BPG | 2.38 | 0.13 | - |
> | L3C | 8.17 | 7.89 | 5M |
> | DLPR | 1.26 | 1.80 | 37M |
> | *Deletang et al.* | *15.13* | *272.05* | 8B |
> | *P²-LLM* | *14.89* | *273.11* | 8.06B(including adaptor)  |
> ---
>
> * *I couldn't find the SCID or BRACS24 datasets anywhere online. Can you provide links to them and (if they exist) citations?*
>
>   **R:** SCID is a standard screen content image dataset (https://onedrive.live.com/?authkey=%21AG8z0EkhES1JQY4&cid=2F2705FEBCB6FF84&id=2F2705FEBCB6FF84%21105&parId=2F2705FEBCB6FF84%21104&action=locate) which is proposed by Ni et al. (ESIM: Edge Similarity for Screen Content Image Quality Assessment, TIP 2017).   BRACS24 is a large-scale histology dataset (https://www.bracs.icar.cnr.it/) which is proposed by Brancati et al. (Brancati N, Anniciello A M, Pati P, et al. Bracs: A dataset for breast carcinoma subtyping in h&e histology images[J]. Database, 2022)

---

> > ### Comment · Reviewer_7XiX · 2025-08-05
> > **Thanks**
> >
> > Thanks for your rebuttal.
> >
> > A couple of points:
> >
> >  - Regarding JPEG-XL, I'm not sure you've fully addressed my point/question. It's not that I'm uncertain how you encoded the JPEG-XL images (although it is helpful to have this detail in the paper). It is that JPEG-XL encodes data other than the raw pixel values. For example it encodes the size of the image. For large images this 'metadata' doesn't contribute much to the compression rate, but for smaller images it is significant. I just want this to be made clear in the paper, that it isn't a perfectly fair comparison, because for your method the compressed bitstream only contains the information required to reconstruct the raw image pixels, whereas in a general purpose format like JXL the compressed bitstream will always need to include some other information.
> >
> >  - Regarding the quote of Mackay, it's nice to cite other sources who also refer to his quote, but the fact that they misrepresent the quote does not make it OK for you to do the same. The two sides, according to MacKay, are _information theory_ and _machine learning_. You can't just substitute whichever other concepts you want and cite him to make your writing sound authoritative (again, even if others have done that it doesn't mean you should). I would suggest either (a) remove the sentence (the paragraph flows fine without it) or if you really want the MacKay quote (b) replace with
> >
> >    > This connection, between machine learning and information theory, dates back to Shannon [CITE a mathematical theory of communication, Shannon 1948] and was famously noted by MacKay (2003), who describes the concepts as "two sides of the same coin".
> >
> >    Then there's no real need to cite the other MacKay quoting papers IMO, just cite Shannon and cite MacKay.

---

> > > ### Comment · Reviewer_7XiX · 2025-08-05
> > > **Sizes of other codecs**
> > >
> > > The table comparing the sizes of the other codecs is helpful and should be added to the paper, but why not include JPEG-XL and BPG? Surely you can find the sizes of the binaries for these codecs and report them, and then report the learned codecs sizes in MB (or GB as appropriate)? The overall point here is that there is still a long way to go before methods like the one presented in your work can compete with non-learned codecs on this kind of portability. I think it's good to be honest about the current limitations of your approach and would improve the paper.

---

> ### Author Response · Authors · 2025-08-04
> **Looking forward to your feedback**
>
> Dear Reviewer 7XiX,
>
> We highly appreciate and acknowledge your very detailed suggestions again. We have addressed each of your concerns in the rebuttal box.
>
> **As the author-reviewer discussion period is ending soon, if there are still remaining concerns, we will do our best to provide more clarifications as soon as possible.**
>
> Once again, we appreciate your time and consideration.
>
> Authors

---

> ### Author Response · Authors · 2025-08-05
> **Response to your follow-up comments**
>
> Dear Reviewer 7XiX,
>
> We thank you for your follow-up comments. We provide point-to-point responses as follows.
>
> ---
>
> **Q1: Comparison clarification over JPEG-XL**
>
>   **R:** Yes, you are right that JPEG-XL encodes data other than the raw pixel values, but our method and other learned image codecs only encode the information required to reconstruct the raw image pixels. To eliminate your concerns, we make several clarifications as follows.
>
> * **Comparison and Testing Standardability.** The in-domain performance of JPEG-XL is directly from the previous SOTA learned codec's paper (DLPR, Deep lossy plus residual coding for lossless and near-lossless image compression, IEEE TPAMI 2024 ). Also, the out-of-distribution performance of JPEG-XL is achieved based on a commonly used library *imagecodecs* (https://github.com/cgohlke/imagecodecs). *All of these attempts aim to avoid unfair comparison by following previous standards.*
>
> *  **We will clarify that JPEG-XL needs to encode other information (e.g., the size of the image) in the real-world compression scenarios, and current comparisons focus on raw pixel values in the revised paper.** Since the function of *imagecodecs* returns the JPEGXL encoded image (https://github.com/cgohlke/imagecodecs/blob/master/imagecodecs/_jpegxl.pyx#L146), it makes current comparisons focus on raw pixel values, which are a little shift from real-world JPEG-XL-like compression scenarios (additionally encoding meta-information). **Therefore, we agree with your suggestion. We will make this point clearer in the revised paper.**
>
> ---
>
> **Q2: The quote of Mackay**
>
>   **R:** We highly appreciate your suggestion! Indeed, your sentence has better readability and is academically rigorous. The original sentence will be replaced by the following sentence:
>
> *This connection, between machine learning and information theory, dates back to Shannon [Shannon, 1948] and was famously noted by MacKay (2003), who describes the concepts as "two sides of the same coin".*
>
> [Shannon, 1948] Shannon, Claude E. "A mathematical theory of communication." The Bell system technical journal 27.3 (1948): 379-423.
>
> [MacKay 2003] David JC MacKay. Information theory, inference and learning algorithms. Cambridge university press, 2003.
>
> ---
> **Q3: Sizes of other codecs**
>
>   **R:** There may be a misunderstanding, i.e., we think you are only interested in the size of learning-based codes in the previous rebuttal. Here, we provide the full table as follows:
>
> | Codec | Encoding Time | Decoding Time | Compressor Size |
> |-------|---------------|---------------|-----------------|
> | JPEG-XL* | 0.73 | 0.08 | [5.98M, 8.91M] |
> | BPG# | 2.38 | 0.13 | [11.3M,12.6M] |
> | L3C | 8.17 | 7.89 | 5M |
> | DLPR | 1.26 | 1.80 | 37M |
> | *Deletang et al.* | *15.13* | *272.05* | 8B |
> | *P²-LLM* | *14.89* | *273.11* | 8.06B(including adaptor)  |
>
> *: The sizes of the binaries of JPEG-XL vary across versions, implementations, and platforms. We provide a range of sizes of the binaries (v.0.11.1) according to the https://github.com/libjxl/libjxl/releases/.
>
> #: The sizes of the binaries of BPG vary across platforms. We provide a range of sizes of the binaries according to the https://bellard.org/bpg/.
>
> **We agree that there is still a long way to go for our method.** Currently, we have tried some smaller models and model quantization techniques to improve the portability (**please see the responses to  Reviewer n2TW, W1**).  Meanwhile, we will discuss this limitation and potential solutions (**e.g., knowledge distillation, model quantization, and pruning**) in the revised paper.
>
> ---
>
> **If there are still remaining concerns, we will do our best to provide more clarifications as soon as possible.**

---

> ### Author Response · Authors · 2025-08-06
> **Thanks for your acknowledgement and looking forward to your follow-up feedback**
>
> Dear Reviewer 7XiX,
>
> Thank you for your acknowledgement. We highly appreciate your thoughtful suggestions that have significantly improved our paper.
>
> **Please let us know if you have any further questions or concerns about our follow-up responses. We would be glad to address them (if applicable) as soon as possible.**
>
> Once again, thanks for your time and consideration. We are looking forward to your follow-up feedback.
>
> Authors

---

### Comment · Area_Chair_igF5 · 2025-08-06

Dear reviewers and authors,

The discussion period is ending soon, so please make sure to finalize the discussion (and make mandatory acknowledgments, if not completed yet). Thank you so much for your active participation!

Best,
AC.

---

### Decision · Program_Chairs · 2025-09-17

**Decision:**

Accept (poster)

**Comment:**

This paper proposes an improved version of the LLM-based lossless image compression algorithm, originally proposed by Deletang et al. (2024). In particular, the paper adopts sequential processing of RGB strings, numerical tokens instead of ASCII embeddings, custom prompts, and LoRA adaptations to substantially advance the state of LLM-based image compression.

On the downside, the technical novelty is not very pronounced, it is still questionable whether LLM-based image compression will be a strong method (due to the computational cost), and the overall presentation of the paper can be improved. In particular, the AC strongly agrees with the comments by the reviewer 7XiX on clarity matters.

Nevertheless, however, all reviewers and AC agree that the paper presents a meaningful improvement over the prior art, on which the future works can build on. Thus I recommend acceptance.